



# Climate Forcing due to Future Ozone Changes: An intercomparison of metrics and methods

William J. Collins[1*], Fiona M. O'Connor[2,3*], Connor R. Barker[4], Rachael E. Byrom[5], Sebastian D. Eastham[6], Øivind Hodnebrog[5], Patrick Jöckel[9], Eloise A. Marais[4], Mariano Mertens[9,10], Gunnar Myhre[5], Matthias Nützel[9,11], Dirk Olivié[7], Ragnhild Bieltvedt Skeie[5], Laura Stecher[9], Larry W. Horowitz[8], Vaishali Naik[8], Gregory Faluvegi[12,13], Ulas Im[14,15], Lee T. Murray[16], Drew Shindell[17], Kostas Tsigaridis[12,13], Nathan Luke Abraham[18,19], James Keeble[20]

[1]Department of Meteorology, University of Reading, Reading, RG6 6 BB, UK
[2] Met Office Hadley Centre, Exeter, United Kingdom
[3] Department of Mathematics & Statistics, Global Systems Institute, University of Exeter, United Kingdom
[4] Department of Geography, University College London, London, United Kingdom
[5] CICERO Center for International Climate Research Oslo, Norway
[6] Faculty of Engineering, Department of Aeronautics, Imperial College London, London, United Kingdom
[7] Norwegian Meteorological Institute, Oslo, Norway
[8] NOAA Geophysical Fluid Dynamics Laboratory, Princeton, NJ, USA
[9] Deutsches Zentrum für Luft- und Raumfahrt, Institut für Physik der Atmosphäre, Oberpfaffenhofen, Germany
[10] Faculty of Aerospace Engineering, Section Operations & Environment, Delft University of Technology, 2629HS, Delft, the Netherlands
[11] Meteorologisches Institut München, Ludwig-Maximilians-Universität München, Munich, Germany
[12] Center for Climate Systems Research, Columbia University, New York, NY, USA
[13] NASA Goddard Institute for Space Studies, New York, NY, USA
[14] Department of Environmental Science, Aarhus University, Roskilde, Denmark
[15] Interdisciplinary Centre for Climate Change, iClimate, Aarhus University, Roskilde, Denmark
[16] Department of Earth and Environmental Sciences, University of Rochester, Rochester, NY, USA
[17] Nicholas School of the Environment, Duke University, Durham, NC, USA
[18] National Centre for Atmospheric Science, United Kingdom
[19] Yusuf Hamied Department of Chemistry, University of Cambridge, Cambridge, United Kingdom
[20] Lancaster Environment Centre, Lancaster University, Lancaster, UK

* These authors contributed equally to this work.

*Correspondence to*: William Collins (w.collins@reading.ac.uk), Fiona O'Connor (F.M.OConnor@exter.ac.uk)

**Abstract.**

We use Earth system models and a chemistry transport model to determine the radiative forcing due to changes in ozone  Three different measures of radiative forcing (instantaneous: IRF, stratospheric-temperature adjusted: SARF, effective: ERF) are compared using both online and offline calculations for the IRF and SARF, and online calculations for the ERF. To isolate the ozone radiative forcing, we configure the model experiments such that only the ozone changes (including respective changes in water vapour, clouds etc.) affect the evolution of the model physics and dynamics. We find robust changes in ozone due to future changes in ozone precursors and ODSs.  These lead to a positive radiative forcing of $0.27\pm0.09$ Wm$^{-2}$ ERF, $0.24 \pm 0.021$ W m$^{-2}$ offline SARF, $0.29 \pm 0.10$ Wm$^{-2}$ online IRF. Increases in ozone lead to an overall decrease in cloud fraction (although



there are increases at some levels). This decrease causes an overall negative adjustment to the radiative forcing (positive in the short-wave (SW), but negative in the long-wave (LW)). Non-cloud adjustments (excluding stratospheric temperature) are positive (both LW and SW). The opposing signs of the cloud and non-cloud adjustments mean the overall adjustment to the SARF is slightly positive.

We find general agreement between models in the impact of the ozone changes on temperature and cloud fractions and

agreement in the signs of the individual adjustment terms when split into SW and LW. However, the overall difference between the ERF and SARF is smaller than the inter-model variability.

## 1 Introduction

Ozone ($O_3$) is an optically active gas, which absorbs and emits longwave (LW) terrestrial infrared (IR) radiation in the 9.6 μm region and absorbs shortwave (SW) solar radiation in the ultraviolet (UV) and visible. Although 90% of ozone is in the

stratosphere and historical changes in the ozone column have been driven by changes in the stratosphere, changes in tropospheric ozone have long been identified as having the larger effect on the radiative forcing. (Fishman et al., 1979; Ramanathan & Dickinson, 1979). This is due to pressure broadening effects on the $O_3$ 9.6 μm band line shape. The Intergovernmental Panel on Climate Change Working Group I (IPCC WGI) 6[th] Assessment Report (AR6) (Forster et al., 2021) based its assessment of ozone radiative forcing on a study by Skeie et al. (2020). Although AR6 did not formally assess the

separate tropospheric and stratospheric contributions to historical forcing, the calculations in Skeie et al. (2020) correspond to a tropospheric ozone radiative forcing of 0.45 Wm[-2] for the period 1750 to 2019. This makes tropospheric ozone the third most important greenhouse gas in terms of historical radiative forcing. The pre-industrial to present-day tropospheric ozone radiative forcing calculation is entirely model based. Uncertainty in our knowledge of this quantity comes from many factors, including inter-model differences in the historical ozone trend, definitions of the radiative forcing and adjustments (see later discussion),

and methodologies of diagnosing radiative forcing. However, since the largest contribution to the uncertainty comes from our lack of knowledge of the pre-industrial emissions of ozone precursors less attention has been given to the methodological uncertainties. The AR6 report assesses a 50% uncertainty in the historical ozone forcing, largely due to the uncertainty in pre-industrial emissions and states "There is also high confidence that this range includes uncertainty due to the adjustments" (Forster et al., 2021).

This study is part of the Tropospheric Ozone Assessment Report Phase II (TOAR-II). In this study we quantify the future ozone radiative forcing from present day to 2050 using consistent changes in ozone depleting substances and ozone precursor emissions between the models. This allows us to focus on how the different definitions of radiative forcing and the use of different methodologies for the calculation affect the quantification of the forcing. We focus on the future period as opposed to historical to provide policy-relevant information on the contribution of tropospheric ozone to future climate change. Previous

studies (Dentener et al., 2006; Gauss et al., 2003; Stevenson et al., 2006, 2013) considered a variety of different scenarios (Special Report on Emissions Scenarios (SRES A2p); Atmospheric Composition Change the European Network of excellence





(ACCENT) current legislation (CLE) and maximum feasible reduction (MFR); and Representative Concentration Pathways (RCPs 2.6, 4.5, 6.0 and 8.5). Here we focus on a Shared Socioeconomic Pathway (SSP3-7.0) with low levels of air pollution-related emission controls (Fujimori et al., 2017).


## 1.1 Radiative forcing

Radiative forcing has proved a useful metric in climate science as it gives a first-order estimate of the potential climatic importance of various forcing mechanisms (Ramaswamy et al., 2019). This follows from the energy balance equation Eq (1):

$\Delta N = \Delta F + \alpha \Delta T$, where $\Delta N$ is the top of atmosphere (TOA) energy imbalance, $\Delta F$ is the applied forcing, $\Delta T$ is the change in global mean surface temperature, and $\alpha$ is the climate feedback parameter (Forster et al., 2021). Eq. (1) implies that applying a forcing will initially push the climate system out of balance ($\Delta N \neq 0$). The climate system will respond with a change in temperature that reduces the magnitude of the energy imbalance with a feedback $\alpha$ until energy balance is restored ($\Delta N = 0$). For a forcing that is constant in time, the system will eventually reach an equilibrium with a temperature change that is directly

proportional to the applied forcing $\Delta T = \Delta F / (-\alpha)$. The feedback parameter $\alpha$ is typically regarded as being approximately independent of the species causing the forcing (Richardson et al., 2019), therefore the radiative forcing is a metric that quantifies the relative temperature effects of perturbations of any species.

### 1.1.1 Definitions of radiative forcing

The simplest definition of radiative forcing is the instantaneous radiative forcing (IRF) which is the change in radiative fluxes

due to the perturbation to atmospheric composition without any other changes. This has historically been calculated at the tropopause since the surface temperatures are strongly correlated with the heating of the surface-troposphere system (Ramanathan et al., 1979). However, since the IPCC 5[th] Assessment Report (AR5) (Myhre et al., 2013) radiative forcing has been defined at TOA. This is the definition that will be used in this paper.

Stratospheric temperatures will respond within a few months to any changes in radiative heating within the stratosphere. It has long been recognized that this stratospheric temperature "adjustment" will affect the long-term climate response to a composition change (Ramanathan et al., 1987). This can be accounted for in the stratosphere by assuming that temperatures adjust to maintain thermal equilibrium with no change in the dynamics (fixed dynamical heating FDH) (Fels et al., 1980). The definition of radiative forcing including these stratospheric-temperature adjustments was used from the First IPCC Assessment

Report (FAR) (Shine et al., 1990) to the 4[th] Assessment report (AR4) (Forster et al., 2007) and was referred to as "radiative forcing" (RF). It was also defined at the tropopause, but since the stratospheric-temperature adjustments bring the stratosphere into radiative balance the net tropopause and TOA fluxes are the same although the partitioning between LW and SW will be





different (Shine et al., 2022). Note that the magnitude and sign of the adjustments depend on whether the IRF is defined at the tropopause or the TOA. AR6 used the terminology "stratospheric-temperature adjusted radiative forcing" (SARF) to clarify
which aspects of the climate system are adjusted. This will be the terminology used in this paper.

A consistent definition of the radiative forcing can also be derived from Eq. (1) since ΔF is just the TOA energy imbalance (ΔN) when ΔT=0. Although this definition appears simple, if the perturbation to the species in question causes subsequent changes in the climate system (that are independent of a change in global mean surface temperature, such as atmospheric
temperatures, water vapour or clouds) the radiative effects of these are implicitly included in the definition of ΔF. This definition was first adopted in the AR5 (Myhre et al., 2013) and referred to as the "effective radiative forcing" (ERF). For the purposes of this paper, we only include changes in physical meteorological fields and are excluding the further effects of ozone on the radiative or microphysical properties of chemically produced greenhouse gases or aerosols, or on the biosphere from the ERF definition. Changes in water vapour due to any thermodynamic response are included. The ERF can be built up from
the IRF by adding in the individual adjustment terms, or by Earth system model (ESM) simulations that implicitly include all the adjustments (see next section).,

Increases in tropospheric ozone cool the lower stratosphere (Checa-Garcia et al., 2018). In the tropopause flux framework this leads to a negative stratospheric-temperature adjustment, reducing the tropopause RF by 20-25% (Checa-Garcia et al., 2018; Shine et al., 1995, 2022) since this reduces the downwards LW flux from the stratosphere to the troposphere. However, in the
TOA framework the stratospheric-temperature adjustment is positive since the stratospheric cooling reduces the upwards LW flux from the stratosphere to space. In Shine et al. (2022) this adds around 80% to the TOA net ozone forcing. Note the net forcing after adjusting the stratospheric temperatures (SARF) must be the same at the tropopause and TOA, the choice of framework only affects the attribution (Shine et al., 2022).

There have been very few studies of meteorological adjustments to tropospheric ozone forcing beyond stratospheric temperatures. Hansen et al. (1997) found that the ratio of temperature warming to (stratospheric-temperature) adjusted radiative forcing varied with the altitude of the ozone change when clouds were included. MacIntosh et al. (2016) suggested that increases in upper tropospheric ozone reduce high cloud and increase low cloud, whereas increases in lower tropospheric
ozone reduce low cloud. Skeie et al. (2020) analysed the adjustments to several meteorological variables from the forcing from historical changes in combined tropospheric and stratospheric ozone using the methodology of Smith et al. (2018). They found a positive stratospheric temperature adjustment due to stratospheric cooling, and a negative tropospheric temperature adjustment (largely offset by a positive water vapour adjustment) due to tropospheric warming. Cloud adjustments were small, but this may have been due to opposite sign responses to tropospheric and stratospheric ozone and compensating changes in
the LW and SW contributions. Since these adjustments were to a combination of historical stratospheric ozone depletion and





tropospheric ozone production, it is not possible to use the Skeie et al. (2020) results to infer adjustments to solely tropospheric ozone changes.

### 1.1.2 Calculations of radiative forcing

In most early studies the radiative forcing of ozone is calculated using radiative transfer models (RTMs; e.g. Ramanathan &
Dickinson, 1979). In these a radiative transfer scheme is run offline from a general circulation model (GCM) perturbing the concentration of ozone. Meteorological parameters (temperature, water vapour, clouds etc.) are supplied as a climatology and so do not adjust to the radiative heating or cooling effects of the ozone (giving an IRF as described above). The exception to this is stratospheric temperatures that are typically adjusted using FDH (Fels et al., 1980) to give a SARF. It is not simple to include further meteorological adjustments in an RTM, hence these do not directly output ERFs. One advantage of using an
offline RTM is that higher-resolution spectral calculations are feasible than would be the case in a GCM (e.g. Myhre et al., 2011). The uncertainty in ozone SARF from RTM calculations was estimated to be around 10% in Stevenson et al. (2013). This uncertainty was mainly due to differences in the radiative code (6%) and specification of clouds (7%), with a smaller contribution (3%) due to the implementation of the FDH.

RTM calculations are still computationally expensive to run many times for many different ozone perturbations. One way of increasing the efficiency of the calculations is to calculate a kernel, i.e. a matrix look-up table of radiative forcing (IRF or SARF) for a unit perturbation of ozone at each vertical level (Skeie et al., 2020). Because each column in an RTM is independent of all the others, a series of layer-by-layer calculations is sufficient to generate a 3D kernel. This kernel can then be multiplied by any 3D ozone perturbation to generate an IRF or SARF. Although the initial calculation of the kernel is
lengthy, once generated it is quick to apply to multiple ozone perturbations. The inaccuracy in using a kernel rather than a full RTM is found to be negligible compared to other uncertainties (Skeie et al., 2020). Figure 1 shows a kernel for the change in TOA SARF based on calculations in Skeie et al. (2020) and further described in section 2.4. The forcing per Dobson Unit (DU) change in ozone is most positive around the tropopause where the atmospheric temperatures are coldest. It is also strongest in the subtropics where the surface temperatures are highest, and the surface albedos are relatively high.




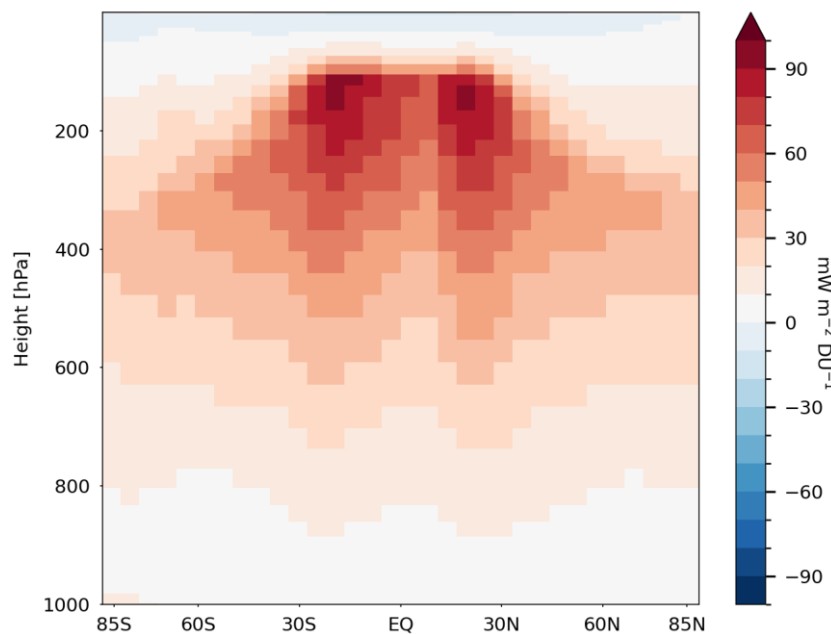

**Figure 1: Radiative efficiencies for ozone SARF in mWm⁻² per DU based on calculations in** Skeie et al. (2020)**.**

RTMs themselves cannot calculate adjustments beyond stratospheric temperatures, but if the adjusted meteorological fields
are provided from a GCM, a full ERF can be calculated. This technique is called a partial radiation perturbation (PRP) and
was originally developed to analyse climate feedbacks (Colman & McAvaney, 1997) but has subsequently been used to analyse
meteorological adjustments (Mülmenstädt et al., 2019; Smith et al., 2018). Not only can the PRP method be used to calculate
a full ERF from an RTM, but if the meteorological changes are imposed one field at a time in the RTM, this method can
decompose the total adjustment into each of its constituent components. Note that due to correlations between the
meteorological changes (e.g. temperature and water vapour) the sum of the individual components is not equal to the total
adjustments after applying the changes cumulatively (Coleman, 2024), although the residuum can be reduced by combining a
forward and backward calculation for each component (Bickel et al., 2020). As described above for the ozone changes, instead
of running the RTM many times it is possible to generate radiative kernels for each of the meteorological fields to generate 3D
(or 2D for surface quantities) radiative sensitivities to changes in temperature, water vapour, cloud, albedo etc (Chung &
Soden, 2015; Myhre et al., 2018; Pendergrass et al., 2018; Smith et al., 2018, 2020). These kernels can then be multiplied by
the 3D (or 2D) meteorological changes to derive the adjustments. This kernel method will suffer from the same inability to
account for correlations as the PRP.



Radiative forcing can also be calculated within GCMs or Earth system models (ESMs). The ERF is defined from equation (1)
as the TOA imbalance when ΔT=0. There are two main methods for calculating this from ESMs. One method is to impose an
abrupt forcing change in a simulation with a coupled ocean and then regress the TOA imbalance against surface temperature
(regression method; Gregory et al., 2004). The ERF is then the TOA imbalance where the regression crosses ΔT=0. A second
method is to fix the surface temperature in an atmosphere-only simulation (Hansen, 2005). Typically for practical model
configuration reasons only the sea-surface temperature is fixed (fSST), however some studies have fixed the land-surface
temperature too (Ackerley & Dommenget, 2016; Andrews et al., 2021b; Shine et al., 2003). In fSST simulations the ERF needs
to be corrected for any temperature change in the land surface (Tang et al., 2019). Both methods suffer from noise due to
interannual variability. This is most pronounced in the regression method (Forster et al., 2016), making it unsuitable for
quantifying the small forcings from changes in tropospheric ozone. The fSST method has less variability but still requires an
integration of over 30 years to reduce the ERF uncertainty to below 0.1 Wm$^{-2}$ (Forster et al., 2016). The internal variability in
the fSST method can be reduced by constraining the winds to prescribed fields (nudging) (Kooperman et al., 2012). However,
this can induce biases in the ERF (Coleman, 2024).

Components of the ERF can be diagnosed within the ESM fSST simulations through extra calls to the radiation scheme.
Typically, these extra calls exclude clouds ("clear sky"), aerosols ("clean sky") or both ("clear-clean sky") (Ghan, 2013). The
difference between "all sky" and "clear sky", or between "clean sky" and "clear-clean sky" can be used to quantify cloud
adjustments. Note though that the "clear sky" or "clear-clean sky" ERF doesn't just remove the cloud adjustments, it also
removes the effects of cloud cover on the instantaneous radiative effect of ozone. The IRF for an ozone perturbation can be
calculated within an ESM by including an additional diagnostic radiation call with a prescribed climatology of ozone
(Dietmüller et al, 2016). The diagnostic radiation calls can be modified to include stratospheric-temperature adjustments
(through FDH) and hence give online ESM calculations of SARF (Stuber et al., 2001; Dietmüller et al., 2016).

In this study we compare IRFs, SARFs and ERFs for tropospheric ozone perturbations for the kernel and online calculations
to compare the consistency or otherwise of the methodologies.

## 1.2 Previous calculations of tropospheric ozone radiative forcing

In this section, we review previous estimates of the pre-industrial to present-day to highlight the diversity in estimates which
is driven not only by methodological diversity but also diversity in the time period of calculations. The IPCC First Assessment
Report (FAR) did not quantify the historical change in tropospheric ozone, but they did calculate radiative efficiencies for
idealised ozone changes. This was 0.02 Wm$^{-2}$ ppb$^{-1}$ based on Hansen (1988) which used equilibrium temperature calculations
from a 1D radiative-convective model. Subsequent reports (see Figure 2) were based on historical ozone changes simulated in
global atmospheric chemistry models with the radiative forcing calculated using offline RTMs (Berntsen et al., 1997; Gauss,
Isaksen, et al., 2006; Hauglustaine et al., 1994; Lelieveld & van Dorland, 1995; Mickley et al., 2001; Skeie et al., 2020;




Stevenson et al., 1998, 2013). The dominant uncertainty in the historical forcing is the lack of knowledge of pre-industrial ozone precursor emissions (Stevenson et al., 2013) rather than uncertainty in the radiative forcing calculations or definitions.

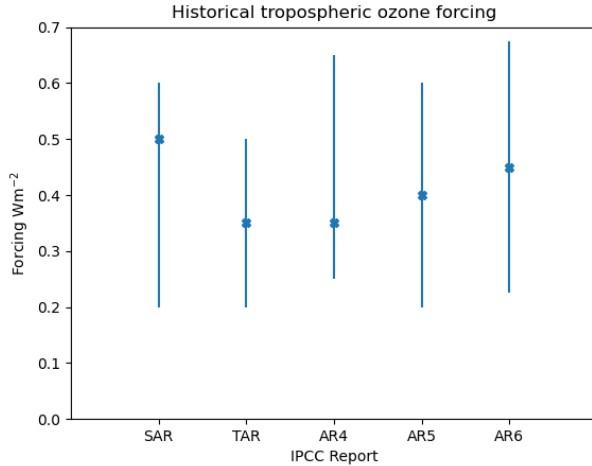


**Figure 2. The tropospheric ozone radiative forcing from "pre-industrial" to "present-day" assessed by the second (SAR) to sixth (AR6) IPCC Assessment Reports. Note the year represented by "present-day" increases throughout the series.**

## 2. Models and Model Simulations

**2.1 Global models**

The global models used in this study include a range of coupled chemistry-climate or Earth System models (CESM2, EMAC, GFDL-ESM4, GISS-E2.1, NorESM2, and UKESM1-0-LL) and the chemical transport model GEOS-Chem. Further details on each of the models are provided in the following.

The Community Earth System Model version 2 (CESM2) used here is described in detail by Danabasoglu et al. (2020). We use the Community Atmosphere Model version 6 with Chemistry (CAM6-Chem) configuration of CESM2 with a nominal 1° horizontal resolution, 32 hybrid sigma-pressure vertical levels and a model top at 2.26 hPa. CAM6 calculates radiative transfer by the correlated k-distribution method via the Rapid Radiative Transfer Model for General circulation models code (RRTMG; Iacono et al. (2008)). Tropospheric and stratospheric chemistry is based on the Model for Ozone and Related chemical Tracers

(MOZART) family of chemical mechanisms with an updated tropospheric chemistry scheme (T1; Emmons et al., 2020), a comprehensive Volatility Basis Set (VBS) parameterisation for the formation of Secondary Organic Aerosols (SOA; Tilmes



et al., 2019) and the MAM4 modal aerosol model (Liu et al., 2016). The tropospheric chemical mechanism includes 151 transported gas-phase species with 65 photolysis reactions and 287 kinetic reactions.

The ECHAM/ Modular Earth Submodel System (MESSy) Atmospheric Chemistry (EMAC) model is comprised of the core atmospheric model ECHAM5 (5th generation European Centre Hamburg general circulation model; Roeckner et al., 2006), and the Modular Earth Submodel system, MESSy, as described by Jöckel et al. (2010). The overall model setup applied in the present study is very similar to the EMAC simulations performed for CCMI-2, which are an update of the setups described by Jöckel et al. (2016). EMAC (MESSy version d2.55.2-5109) is applied in the T42L90MA-resolution, i.e. with a spherical

truncation of T42 (corresponding to a quadratic Gaussian grid of approx. 2.8° by 2.8° in latitude and longitude) with 90 vertical hybrid pressure levels up to 0.01 hPa. For the chemical kinetics, the submodel MECCA (Module Efficiently Calculating the Chemistry of the Atmosphere; Sander et al., 2019) is used. The chemical mechanism considers the basic gas-phase chemistry of ozone, methane, and odd nitrogen. Alkanes and alkenes are included up to C4. Alkynes and aromatics are not considered in our mechanism. Halogen chemistry includes bromine and chlorine species. For the chemistry of isoprene plus a few selected

non-methane hydrocarbons (NMHCs), we used version 1 of the Mainz Isoprene Mechanism (MIM1) based on Pöschl et al. (2000). In total, 265 gas phase, 82 photolysis and 12 heterogeneous reactions for 204 chemical species are considered. Radiation is calculated based on the ECHAM5 radiation scheme, i.e. RRTM in the LW and the Fouquart & Bonnel (1980) scheme in the SW plus FUBrad (Kunze et al., 2014), as described by Dietmüller et al. (2016) with the MESSy submodel RAD, which allows for multiple diagnostic radiation calls (Dietmüller et al., 2016; Nützel et al., 2024).


The GFDL Earth System Model version 4 (GFDL-ESM4) is documented by Dunne et al. (2020), using the atmospheric component AM4.1 as described by Horowitz et al. (2020). The base configuration used here is the same as that used for Phase 6 of the Coupled Model Intercomparison Project (CMIP6; Eyring et al., 2016). Briefly, the model has a horizontal resolution of approximately 100 km (~ 1° latitude by 1° longitude), using the GFDL Finite-Volume Cubed-Sphere dynamical core (L.

M. Harris & Lin, 2013; Putman & Lin, 2007), with 49 hybrid sigma-pressure vertical levels extending from the surface to 1 Pa (~80 km). AM4.1 includes interactive tropospheric and stratospheric gas-phase and aerosol chemistry. The combined tropospheric and stratospheric chemistry scheme includes 18 prognostic (transported) bulk aerosol tracers, 58 prognostic gas-phase tracers, five prognostic idealised tracers, and 40 diagnostic (non-transported) chemical tracers, with 43 photolysis reactions, 190 gas-phase kinetic reactions, and 15 heterogeneous reactions. The tropospheric chemistry includes reactions of

the $NO_x$–$HO_x$–$O_x$–CO–$CH_4$ system and oxidation schemes for other non-methane volatile organic compounds. The stratospheric chemistry accounts for the major ozone loss cycles ($O_x$, $HO_x$, $NO_x$, $ClO_x$, and $BrO_x$) and heterogeneous reactions on ice and nitric acid trihydrate polar stratospheric clouds (PSCs) and in liquid ternary solution (LTS) aerosols. Emissions of biogenic volatile organic compounds (VOCs), dust, sea salt, dimethyl sulphide (DMS), marine organic aerosols, and lightning $NO_x$ are calculated interactively as a function of model meteorology. The radiation scheme in GFDL-ESM4 has been

comprehensively updated in the SW (Exponential Sum Fit scheme; ESF) and LW (Simplified Exchange Approximation



scheme; SEA); a full description can be found in Zhao et al. (2018). The land component (LM4.1) of the model uses an interactive dynamic vegetation scheme to simulate vegetation dynamics (Shevliakova et al., 2024).

This study used version 2.1 of the GISS ModelE Earth system model, ModelE2.1 (GISS-E2.1), as described and evaluated for the present day (Kelley et al., 2020), historical period (Miller et al., 2021) and in future projections (Nazarenko et al., 2022). The horizontal and vertical resolution of the atmosphere in ModelE2.1 is 2° in latitude by 2.5° in longitude with 40 vertical layers from the surface to 0.1 hPa. The radiative transfer model is described by Lacis & Oinas (1991). Tropospheric chemistry (Shindell et al., 2001, 2003) includes the inorganic chemistry of reactive oxygen ($O_x$), nitrogen ($NO_x$), and hydrogen ($HO_x$) families as well as carbon monoxide (CO), and the organic chemistry of methane and higher hydrocarbons using a modified Carbon Bond Mechanism version 4 (CBM4) scheme (Gery et al., 1989). The stratospheric chemistry includes chlorine and bromine chemistry together with polar stratospheric clouds (Shindell et al., 2006). The original chemical scheme has been updated and evaluated since its first implementation (Shindell et al., 2013) and has continued to be updated since (Rivera et al., 2024). ModelE2.1 includes multiple aerosol schemes (Bauer et al., 2020): One-Moment Aerosols (OMA), which is the scheme used here, is fully interactive within ModelE2.1 in terms of emissions, chemistry, transport, removal, and climate. Aerosol-radiation interactions (ARIs) and aerosol–cloud interactions (ACIs) are calculated within the radiation and cloud schemes, where the size-dependent scattering properties of clouds and aerosols are computed from Mie scattering. Apart from swelling with water, there is no internal mixing in OMA radiative calculations - all aerosols are regarded as externally mixed.

The Norwegian Earth System Model used in this study is based on the standard version of NorESM2 (Seland et al., 2020) but has been extended with comprehensive atmospheric chemistry. In this new version of NorESM2 the atmospheric aerosol scheme (Kirkevåg et al., 2018) has been coupled with the gas-phase description of the TS1 troposphere-stratosphere chemistry scheme (Emmons et al., 2020) of CESM2. This new version now describes a total of 210 species (aerosol and gas-phase) and is currently under evaluation. As in the standard version, natural emissions of dust, isoprene and monoterpenes from land and of primary organic matter, sea-salt and DMS from the ocean are calculated interactively. The atmosphere and land components of NorESM2 used here have the same horizontal resolution of 1.9°x2.5°, and the atmosphere contains 32 levels (up to 2.26 hPa). NorESM2 is based on CESM2 (it also uses the RRTMG radiative transfer scheme) but differs in quite some aspects. NorESM2 contains a different ocean model, a different atmospheric aerosol module (including its interactions with cloud and radiation), and modifications in the formulation of local dry and moist energy conservation, in local and global angular momentum conservation, and in the computations for deep convection and air–sea fluxes. The surface components of NorESM2 also have minor changes in their albedo calculations.

The UK's Earth System Model used in this study is based on UKESM1-0-LL (Sellar et al., 2019), as used in CMIP6. It has a horizontal resolution of 1.25°×1.875°, with 85 hybrid height levels from the surface up to the model lid at 85 km. By default, UKESM1-0-LL includes an interactive stratosphere-troposphere chemistry scheme called StratTropv1.0 (T Archibald et al.,



2020) from the United Kingdom Chemistry and Aerosol (UKCA; Morgenstern et al., 2009; O'Connor et al., 2014) model. However, near-global total column ozone was biased high relative to other CMIP6 models throughout the historical period (Keeble et al., 2021) and the stratospheric ozone response to increasing concentrations of ozone-depleting substances was too strongly negative (Morgenstern et al., 2020), resulting in UKESM1-0-LL having a negative present-day ozone forcing (Skeie et al., 2020). Here, the model simulations use UKESM1-0-LL but include an update to the chemistry scheme, i.e., StratTropv2.0 (Keeble et al., 2024; In preparation). In comparison with StratTropv1.0, StratTropv2.0 includes an extension to the stratospheric heterogeneous chemistry to incorporate bromine species as well as updates to bimolecular reaction rate coefficients, the treatment of the top boundary, and photolysis inputs. The radiation scheme used in UKESM1-0-LL is SOCRATES (Suite Of Community RAdiative Transfer codes based on Edwards and Slingo) which treats incoming solar radiation and thermal radiation in six SW bands and nine LW bands, respectively. A full description of SOCRATES and the physical climate model on which UKESM1-0-LL was built can be found in (Walters et al., 2019). Apart from the treatment of trace gas chemistry, the configuration of UKESM1-0-LL used here is identical to that used in CMIP6.

In contrast to the other models in this study, GEOS-Chem (http://www.geos-chem.org) is a 3-D global chemistry-transport model (CTM). The model is driven by meteorological fields archived at high temporal resolution from either reanalyses or free-running GCMs. The computational power that is not used to resolve the equations of motion is therefore applied instead to include additional chemical species and more complex chemical mechanisms. Here, version 14.4.2 of GEOS-Chem is used, whose chemistry mechanism includes 263 gas-phase and 30 condensed-phase species, undergoing 624 gas-phase, 113 heterogeneous, and 157 photolysis reactions in a unified mechanism from the surface to the mesopause, with a linearized mechanism applied in the mesosphere (e.g., Bates et al. (2024), and ref. therein). Unlike the other simulations used in this study, the mechanism includes a comprehensive tropospheric halogen chemistry mechanism for the $Br_y$, $Cl_y$, and $I_y$ families, which strongly influences the tropospheric ozone budget (X. Wang et al., 2021). The tropospheric ozone simulation was recently evaluated against observations by H. Wang et al. (2022). The source of NOx from lightning is described by Murray et al. (2012) and the source from soil microbes is described by Hudman et al. (2012). Biogenic VOC emissions are from version 2.1 of the Model of Emissions of Gases and Aerosols from Nature (MEGAN) inventory of Guenther et al. (2012) as implemented by Hu et al. (2015). The RRTMG radiative transfer scheme is embedded within GEOS-Chem, enabling calculation of online IRF at the surface, tropopause, and TOA from greenhouse gases and aerosol particles (Heald et al.,2014).

## 2.2 Model simulations

The protocol for the model simulations carried out here is as follows.

As described in Section 1.1.2 and following recommendations from Forster et al. (2016) for the quantification of ERFs, the model simulations conducted here are atmosphere-only, with prescribed sea surface temperatures (SSTs) and sea ice cover





(SIC). The control experiment (called *pdClim-control*) is a time-slice simulation for the year 2015, using prescribed climatologies for SSTs and SIC appropriate for the year 2015. Long-lived greenhouse gas concentrations for the year 2015,

including those for ozone-depleting substances (ODSs), nitrous oxide, and methane, are taken from the Shared Socioeconomic Pathway 3-7.0 (SSP3-7.0; Meinshausen et al., 2020). Other boundary conditions, such as ocean concentrations of DMS, etc. are also prescribed as climatologies appropriate for the present day. Emissions of non-methane ozone precursor gases, aerosols, and aerosol precursors are prescribed as annually repeating emissions, using distributions and global annual totals from SSP3-7.0 for the year 2015, as prescribed for CMIP6. For those models that are free running (prescribed SST/SIC), the control

simulation was run for 50-70 years, allowing 30-40 years to be available for analysis following spin up to reduce the effects of internal model variability. For those models that use specified dynamics (see below), the model control simulation was typically shorter in length following spin up, i.e., 5-12 years in length.

The perturbation simulation (called *pdClim-2050ssp370-radO3*) is identical to that of the control, except that methane and

ODS concentrations and all short-lived climate forcer emissions, as seen by the chemistry are prescribed, using year-2050 values from SSP3-7.0. Although greenhouse gases, ozone precursors and aerosol emissions are perturbed here relative to the control, only perturbations to ozone itself can affect the top-of-atmosphere (TOA) radiative fluxes. The models' respective radiation and cloud schemes continue to see year-2015 atmospheric concentrations, except for ozone. As a result, any difference in the TOA radiative fluxes between *pdClim-control* and *pdClim-2050ssp370-radO3* is solely due to 2015-to-2050

changes in ozone (driven by changes in precursor emissions and ODS and GHG concentrations) and any resulting rapid adjustments.

As the standard perturbation simulation also incorporates changes to stratospheric ozone due to the expected reductions in ODSs (WMO, 2022), an additional perturbation simulation, *pdClim-2050ssp370fODS-radO3*, was performed by EMAC,

GFDL-ESM4, and UKESM1-0-LL. This additional perturbation simulation is identical to the standard perturbation simulation, *pdClim-2050ssp370-radO3*, except that ODSs were held at year-2015 values in the chemistry. Hence, this perturbation, which will be referred to as fODS-perturbation in the following and is designed to remove the effect of changing ODSs and connected changes in stratospheric ozone. While the *pdClim-2050ssp370-radO3* minus *pdClim-2050ssp370fODS-radO3* ozone difference will be *solely* attributable to changes in ODSs, the focus of analysis will be on tropospheric differences between the

fODS-perturbation and the present day (*pdClim-2050ssp370fODS-radO3* minus *pdClim-control*). Nitrous oxide ($N_2O$) may be an important factor with respect to ozone depletion in the 21st century, but the ozone response in the troposphere will be largely driven by changes in tropospheric ozone precursors in SSP3-7.0. A summary of the experimental setup for these 3 simulations can be found in Table 1 and Table 2 lists the length of spin up and the number of years available for analysis from the participating models.






| Experiment Name | SSTs/SI | WMGHGs as seen by radiation | N2O as seen by chemistry | ODSs as seen by chemistry | CH4 as seen by chemistry | VOC, CO & NOx as seen by chemistry | Aerosols, as seen by radiation & clouds | Aerosols as seen by chemistry |
|---|---|---|---|---|---|---|---|---|
| *pdClim-control* | Year 2015 | Year 2015 | Year 2015 | Year 2015 | Year 2015 | Year 2015 | Year 2015 | Year 2015 |
| *pdClim-2050ssp370-radO3* | Year 2015 | Year 2015 | Year 2050 | Year 2050 | Year 2050 | Year 2050 | Year 2015 | Year 2050 |
| *pdClim-2050ssp370fODS-radO3* | Year 2015 | Year 2015 | Year 2050 | Year 2015 | Year 2050 | Year 2050 | Year 2015 | Year 2050 |

**Table 1: List of the atmosphere-only (fSST) experiments carried out in this study to quantify year-2050 ozone effective radiative forcing relative to year-2015, due to changes in well-mixed greenhouse gases (WMGHGs) and non-methane ozone precursors.**

In some cases, the protocol could not be implemented in a straightforward way. As a result, some bespoke model changes were
made and are documented in the following.

In NorESM2 and CESM2, we were not able to implement the experimental protocol for the year-2050 perturbation simulation in all its aspects. For GHGs and ODSs, the model radiation scheme was successfully able to see GHGs/ODSs representative for 2015, while the model chemistry included GHGs/ODSs representative of the year 2050. However, this separation was not
so feasible for both aerosols and methane-driven production of water vapour. In principle, in the perturbation simulation, the radiation and clouds (through microphysics) should see year-2015 aerosols, whereas the chemistry should see year 2050 aerosol surface densities for heterogeneous reactions.

Therefore, to obtain the ozone ERF in CESM2 we perform one further control experiment (called *pdClim-control-fixO3*) and
one further perturbation experiment (called *pdClim-2050ssp370fixO3-radO3*). These are identical to CESM2's *pdClim-control* and *pdClim-2050ssp370-radO3* in all aspects apart from the prescription of an O$_3$ climatology field that is seen only by the radiation scheme. In both cases, this climatology represents O$_3$ concentrations from the year 2010 as zonally-averaged 5-day fields. As with *pdClim-control* and *pdClim-2050ssp370-radO3*, these additional experiments are run for 50 years, with the last 40 years used for analysis. Together, these simulations allow us to calculate the radiative and microphysical (cloud) impact of



2050 aerosols and the radiative effect of $CH_4$ driven changes in stratospheric water vapour and isolate the $O_3$ ERF by differencing the two experiment sets as: (*pdClim-2050ssp370-radO3 – pdClim-control*) - (*pdClim-2050ssp370fixO3-radO3 – pdClim-fixO3*), as illustrated in Fig. S1 of the Supplementary Material.

  A similar approach is used in NorESM2, whereby the 3D ozone climatology generated in *pdClim-control* and *pdClim-*

*2050ssp370-radO3* (from years 21-40) are used as input in two further simulations, R2 and P2, respectively. R2 is identical to NorESM2's *pdClim-control* in all aspects apart from the prescription of an $O_3$ climatology field that is seen only by the radiation scheme that is representative of $O_3$ distribution for the year 2015. Likewise, P2 is identical to *pdClim-control* apart from the prescription of $O_3$ climatology representative of the year 2050. Differencing P2 and R2 then allows us to isolate the $O_3$ ERF in NorESM2.


  In EMAC, we use nudging or "specified dynamics", i.e. Newtonian relaxation towards ERA-5 data (Hersbach et al., 2020), so that the reference and the perturbation simulation have the same meteorology. The relaxation is applied in spectral space for the prognostic variables of divergence, vorticity, temperature, and the (logarithm of the) surface pressure. The mean

temperature (i.e. wave zero) is not nudged (see Jöckel et al. (2016) for more details). Therefore, only 5 simulation years are necessary for the analyses. Furthermore, we use a slight variation of the quasi Chemistry-Transport-Model mode described by Deckert et al. (2011). This mode enables a decoupling of dynamics and chemistry. For this we performed one simulation with full coupling of dynamics and chemistry but otherwise the same setup as the *pdClim-control* simulation (called *pre*). The setup uses prescribed climatologies for tropospheric and stratospheric aerosol for the radiation calculation and for the heterogeneous

chemistry on particle surfaces. The aerosol climatologies used for heterogeneous chemistry stem from the model simulations described by Righi et al. (2023) with different climatologies for the 2015 simulation (SSP2-4.5) and the 2050 simulations (SSP3-7.0). For the radiation calculation in the *pdClim-control, pdClim-2050ssp370-radO3* and *pdClim-2050ssp370fODS-radO3* simulation from EMAC, all radiative active trace gases, except for ozone and water vapour, are then prescribed from monthly mean transient files from this *pre*-simulation. Moreover, we used monthly mean OH values from this simulation with

the $CH_4$ lower boundary condition from *pdClim-control* for a parameterized methane oxidation scheme as a source for stratospheric water vapour in all simulations via the $CH_4$ submodel (Winterstein & Jöckel, 2021).

  In GFDL-ESM4, the chemical production of water vapour, including from the oxidation of methane and hydrogen, is disabled and replaced by nudging stratospheric water vapour to an analytic approximation of the climatology retrieved by the HALogen

Occultation Experiment (HALOE), as described by L. Harris et al. (2021). The aerosol concentrations used in the radiation and aerosol activation schemes are prescribed externally and are identical in the control and perturbation simulations. The aerosol concentrations used were scaled from the aerosol concentration forcing fields provided for CMIP5, rescaled globally to approximately match the global mean aerosol optical depths and burdens simulated by GFDL-ESM4. In addition, the ability




to add an additional diagnostic radiative transfer call with ozone set to zero throughout the atmosphere was added to GFDL-
ESM4 to allow calculations of ozone's direct radiative effect.

Two sets of simulations were carried out with the GISS-E2.1 model – free-running (FR) and nudged. No additional bespoke
changes were made to the GISS-E2.1 FR simulations, denoted as GISS-E2.1_FR hereafter. In GISS-E2.1 with nudging – called
GISS-E2.1_nudged hereafter – horizontal winds only were nudged towards year-2015 of a MERRA2 three-hourly dataset
(Gelaro et al., 2017) and the simulations were run for more years than for the EMAC nudged model (Table 2). For the control
simulation, SSP2-4.5 2015 emissions were used, though Shared Socioeconomic Pathway emissions were similar among
pathways for this year, as they were harmonized to meet historical emissions in 2015 (Gidden et al., 2018). The perturbation
simulation used SSP3-7.0 year-2050 emissions. Sea surface conditions and vegetation were prescribed as 2015 values, as
prepared for CMIP6 (Kelley et al., 2020), rather than as a climatology. A few alterations were made to the model code to
conform closely to the protocol. With standard code, the radiative transfer calculation experiences chemical ozone changes
without experiencing methane or aerosol changes. However, the chemical change to water vapour reaching the radiation code
is normally changed in unison with ozone. In the simulations here, this was decoupled, such that the water vapour changes due
to chemistry did not affect the radiation. An alteration was also needed to allow the perturbation simulation to use year-2050
greenhouse gases ($CH_4$, CFCs, $N_2O$) in the chemistry while maintaining year-2015 values in the radiation calculations.


In UKESM1-0-LL, stratospheric water vapour production from methane oxidation, as documented in Archibald et al. (2020),
was deactivated. In its place, a parameterization for methane oxidation independent of the chemistry scheme was activated, in
which the methane mixing ratio was implicit and derived from the assumption that 2 [CH4] + [H2O] = 3.75 ppm throughout
the stratosphere. In this way, from a radiative perspective, stratospheric water vapour in *pdClim-control* and *pdClim-*
*2050ssp370-radO3* was set close to a value of 3.75 ppm. In addition, an extra diagnostic call to the SOCRATES radiative
transfer scheme with ozone set to a prescribed climatology was added to allow the calculation of ozone's direct radiative effect.

**2.3 Online Double Radiation Calls for Ozone**

In addition to the diagnosis of TOA radiative fluxes for the purpose of quantifying ERF, some models included a diagnostic
call to the radiation scheme with a reference ozone field. Analogous to the Ghan method for aerosols (Ghan, 2013), differences
between radiative fluxes from the diagnostic and prognostic radiation calls between the control and perturbation experiments
enable an ozone IRF to be calculated. In some models (e.g., EMAC), the diagnostic call included a stratospheric temperature
difference relative to the prognostic call; in this case, an ozone SARF can be calculated. In other models (e.g., GEOS-Chem),
the online double radiation call for ozone was the only online means of diagnosing an ozone forcing metric. Table 2 lists which
models included online double radiation calls for ozone and for which metric.



## 2.4 Offline Ozone Radiative Kernel Method

To calculate TOA SARF, a top of atmosphere version of the radiative kernel for stratospheric adjusted radiative forcing (SARF) at the tropopause (Skeie et al., 2020) is used. The monthly mean ozone mixing ratios in *pdClim-ssp370-radO3* and *pdClim-control* are regridded horizontally and vertically to the resolution of the radiative kernel (T21 (~5.6° at the equator) and 60

vertical levels from the surface and up to 0.1 hPa). The radiative kernels are monthly 3-dimensional fields of RF at top of atmosphere per DU change for SW (clear sky all sky), LW (clear sky, all sky) and LW all sky including stratospheric temperature adjustments (Fig. S2). Net SARF are calculated as the sum of SW all sky and LW all sky adjusted. For each year in the model simulations, the kernel is multiplied by the difference in ozone mole fractions between the two experiments converted to DU using common meteorological fields native to the radiative kernel.


In addition to the calculation of TOA SARF for the total ozone changes, separate calculations are also performed for tropospheric ozone changes, where the tropopause is defined for each month based on the ozone mole fraction ($< 150$ nmol mol$^{-1}$) in *pdClim-control*.






| Model | Nudged | Free-running | Spin up | Analysis period | Radiative Transfer Scheme | Online IRF | Online SARF | Online ERF | Offline SARF |
|-------|--------|--------------|---------|-----------------|----------------------------|------------|-------------|------------|--------------|
| CESM2 | No | Yes | 10 years | 40 years | RRTMG | No | No | Yes | Yes |
| EMAC | Yes | No | 2 years* | 5 years | MESSy submodel RAD (see text) | Yes | Yes | Yes | Yes |
| GEOS-Chem | Yes | No | 9 years | 5 years | RRTMG | Yes | No | No | Yes |
| GFDL-ESM4 | No | Yes | 30 years | 40 years | GFDL (SEA-ESF) | Yes | No | Yes | Yes |
| GISS-E2.1_FR | No | Yes | 10 years | 30 years | GISS | Yes | No | No | Yes |
| GISS-E2.1_nudged | Yes | No | 4 years | 12 years | GISS | Yes | No | Yes | Yes |
| NorESM2 | No | Yes | 10 years | 30 years | RRTMG | No | No | Yes | Yes |
| UKESM1-0-LL | No | Yes | 30 years | 40 years | SOCRATES | Yes | No | Yes | Yes |

**Table 2: List of participating global models in quantifying future ozone forcing. Table also includes the length of the model spin up, the number of years available for analysis, and which ozone forcing calculations are available from the different models. *In the EMAC model, an additional spin up of 10 years was undertaken in the *pdClim-2050ssp370fODS-radO3* simulation.**

## 3. Results

**3.1 Modelled ozone changes**

From the experimental setup described above, year-2015 and year-2050 ensemble mean climatologies for total column ozone (TCO) are shown in Figure 3, with results from the individual models shown in Figure S3 of the Supplementary Material. At the present day and based on 7 of the 8 participating models, the multi-model global mean TCO is 298.3 ± 8.3 DU, with



minimum values in the tropics and southern high latitudes and maximum values in the northern mid-to-high latitudes.

Following SSP3-7.0, global mean TCO increases to 310.5 ± 10.4 DU in 2050 in the ensemble mean, with the largest increase occurring in the southern high latitudes. From a multi-model perspective, this represents a global mean TCO increase of 12.2 ± 5.2 DU in response to reduced ODSs combined with increases in nitrous oxide, carbon dioxide, methane and non-methane ozone precursors. Looking at the individual model responses, the increase in global mean TCO is in the range of 2.1 DU (GISS-E2.1_nudged) to 19.6 DU (UKESM1-0-LL). Most models also show increases in TCO in all regions of the globe. While

GISS-E2.1_nudged shows a weak increase in global mean TCO in 2050 (Figure S3), decreases in the northern high latitudes and particularly in the southern high latitudes are evident. This anomalous response may be due to that model's implementation of nudging (Orbe et al., 2020) and is contrary to the scientific understanding of stratospheric ozone recovery and CMIP6 projections (Keeble et al., 2021). As a result, GISS-E2.1_nudged was omitted from the ensemble mean in Figure 3 and all ensemble means hereafter. This seven-model ensemble is hereafter referred to as the TOAR-RF ensemble. The range for the

increase in global mean TCO from the remaining models is 4.3-19.6 DU.





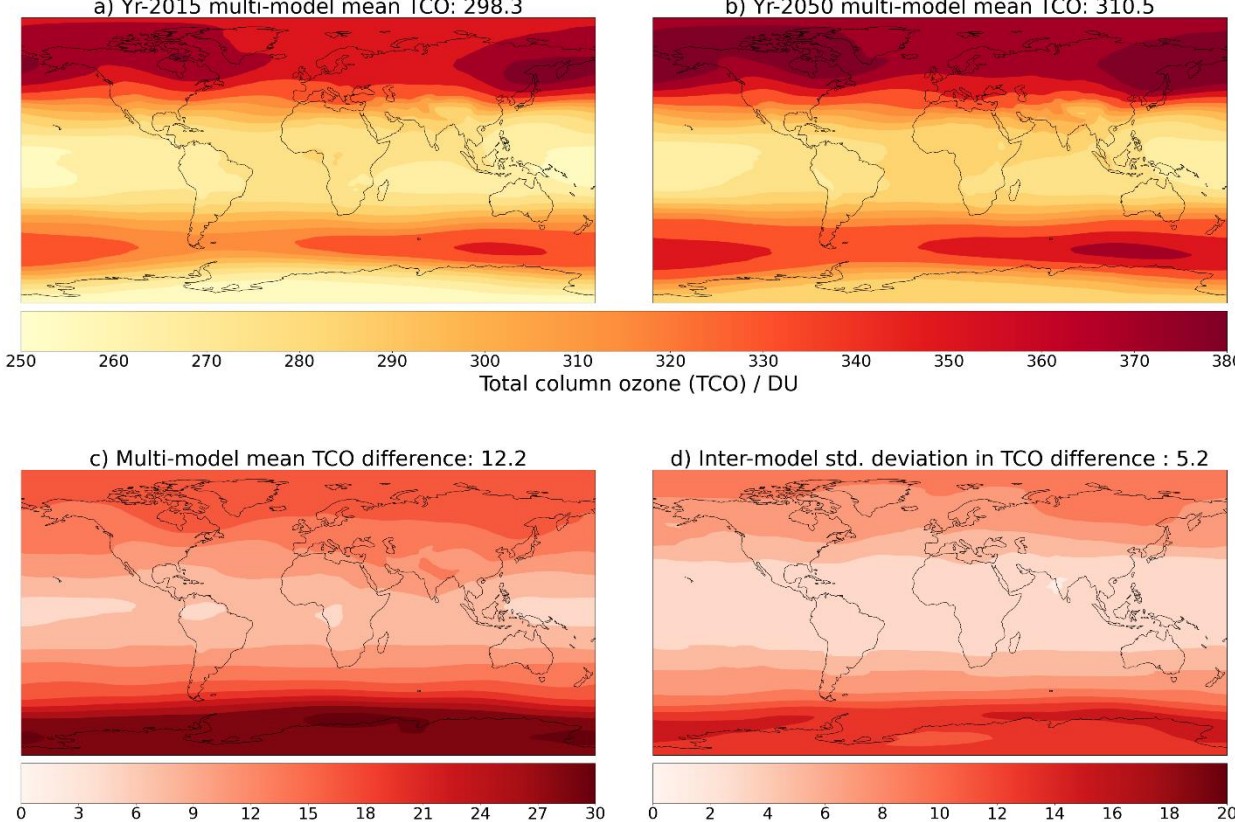

**Figure 3: Multi-model mean climatologies of total column ozone (TCO) in Dobson Units (DU) for a) present day (year 2015) and b) future (year 2050) following the SSP3-7.0 scenario, c) the multi-model mean difference between the climatologies (year-2050 minus year-2015), and d) the inter-model standard deviation about the multi-model mean difference. Models included in the multi-model means are CESM2, EMAC, GEOS-Chem, GFDL-ESM4, GISS-E2.1_FR, NorESM2, and UKESM1-0-LL. Global ensemble mean values are shown above each panel.**

By using the definition of the tropopause based on the ozone mole fraction ($< 150$ nmol mol$^{-1}$) in *pdClim-control* and the same 7-member ensemble as for TCO, year-2015 and year-2050 climatologies for tropospheric column ozone (TrCO) from the ensemble mean and from all of the individual models are shown in Figures 4 and S4, respectively. In the present day, TrCO shows multi-model global mean values of $36.2 \pm 1.1$ DU, with maximum values in the sub-tropics and northern mid-latitudes and minimum tropospheric column amounts over the tropical Pacific and southern high latitudes. Following SSP3-7.0, the multi-model global mean TrCO increases to $40.5 \pm 1.1$ DU in 2050, representing a multi-model global mean increase of $4.3 \pm 1.0$ DU. This indicates that of the TCO changes shown in Figure 3, $39 \pm 14$ % of the increase occurs within the troposphere.



Regionally, the largest increase in TrCO occurs over the middle East, India, and southeast Asia and the largest inter-model spread in the TrCO increase is in the southern high latitudes. Looking at the individual models, present-day global mean TrCO ranges from 35.0 DU in GFDL-ESM4 to 39.7 DU in GISS-E2.1_nudged. While most models broadly agree on the spatial
distribution in the present day, maximum column amounts extend further into the northern high latitudes in the GISS-E2.1 model simulations, and they also show the deepest minimum over the tropical Pacific. In 2050, following SSP3-7.0, TrCO increases globally in all models, with global mean values in the range of 39.1-42.2 DU. However, the increases are more regionally confined to the northern hemisphere sub-tropics in the GISS-E2.1 model simulations whereas the other models show that substantial increases also occur elsewhere (e.g., high latitudes) in response to the changes in well-mixed greenhouse
gases and ozone precursors in SSP3-7.0.

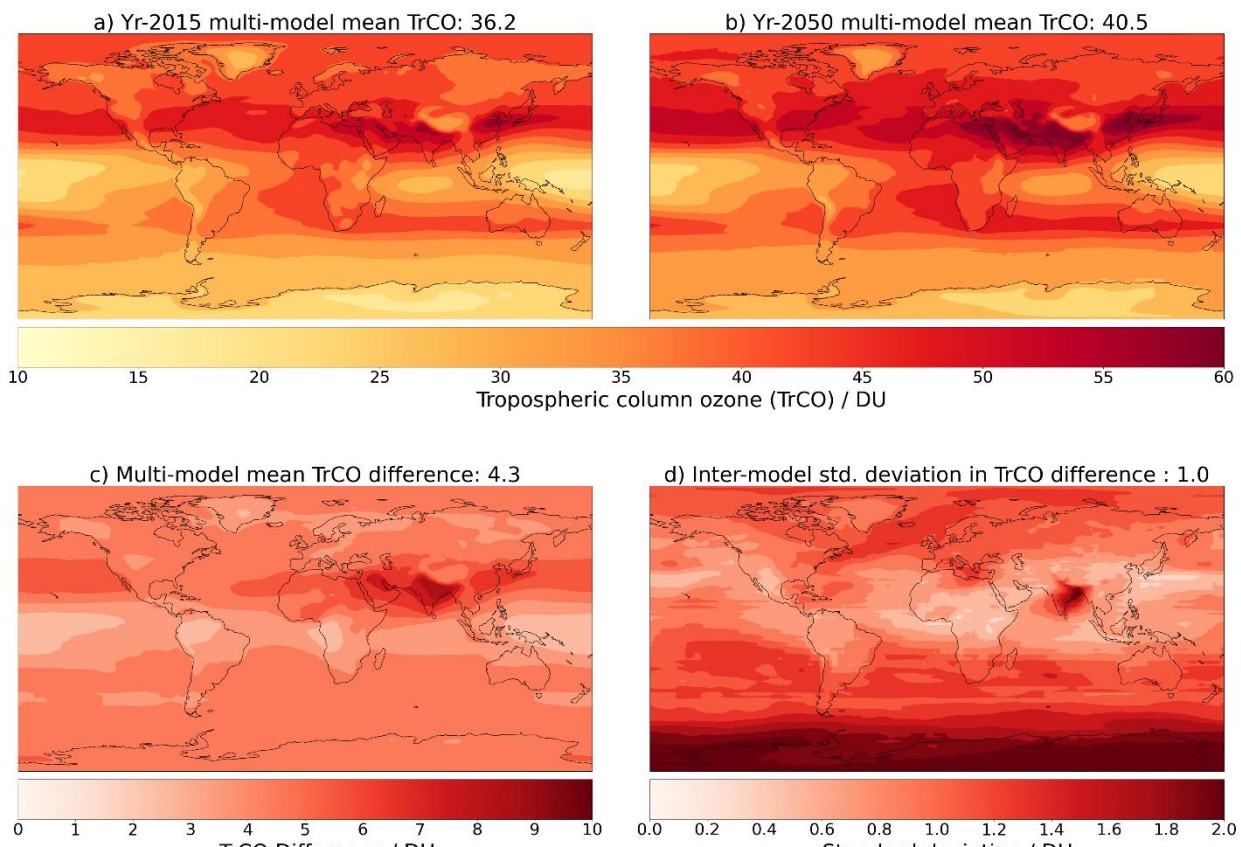

**Figure 4: Multi-model mean climatologies of tropospheric column ozone (TrCO) in Dobson Units (DU) for a) present-day (year-2015) and b) future (year 2050) global distributions, c) the multi-model mean difference between the climatologies (year-2050 minus year-2015), and d) the inter-model standard deviation about the multi-model mean difference. Models included in the multi-model**
**means are the same as in Fig. 3. Global ensemble mean values are shown above each panel.**





Figure 5 shows the TOAR-RF 7-member ensemble-mean zonal-mean ozone climatology for the present-day (year-2015) and future (year-2050), along with the difference between them. Climatologies for each individual model are shown in Fig. S5. In the 7-model ensemble mean, future zonal mean ozone increases almost everywhere from present day values following SSP3-7.0. The only exception to this is in the tropical and northern hemisphere stratosphere, centred around 10 hPa, where a reduction in zonal mean ozone of less than 1 % occurs. The largest increase occurs in the southern high latitudes between 60 and 90 °S and 300-80 hPa where zonal mean increases of greater than 20 % are evident. However, the variability in the modelled response in this region also reaches a maximum, with the inter-model standard deviation in the zonal mean relative difference being greater than 10 %. In the troposphere, zonal mean ozone increases in all regions in 2050. Increases in tropospheric ozone precursors drive increases of up to 14 %, with the largest increase occurring through the depth of the troposphere in the northern hemisphere tropics and sub-tropics. The inter-model standard deviation in the tropospheric zonal mean ozone increase is typically between 2 and 4 %.

Looking at the individual model zonal mean climatologies and responses (Fig. S5), the models indicate that the largest relative changes occur in the southern high-latitude upper troposphere and lower stratosphere. Apart from the GISS-E2.1_nudged simulation, there are weaker positive and negative changes aloft and there is evidence of secondary peak increases in the extra-tropical stratosphere (between 1 and 10 hPa) for those models with a higher model lid. Although GISS-E2.1_nudged shows the strongest increases in this region, the negative TCO changes in the future (Fig. S3) result from reductions in ozone of 5-30 % throughout the lower stratosphere, with the largest reductions occurring in the southern high latitudes.



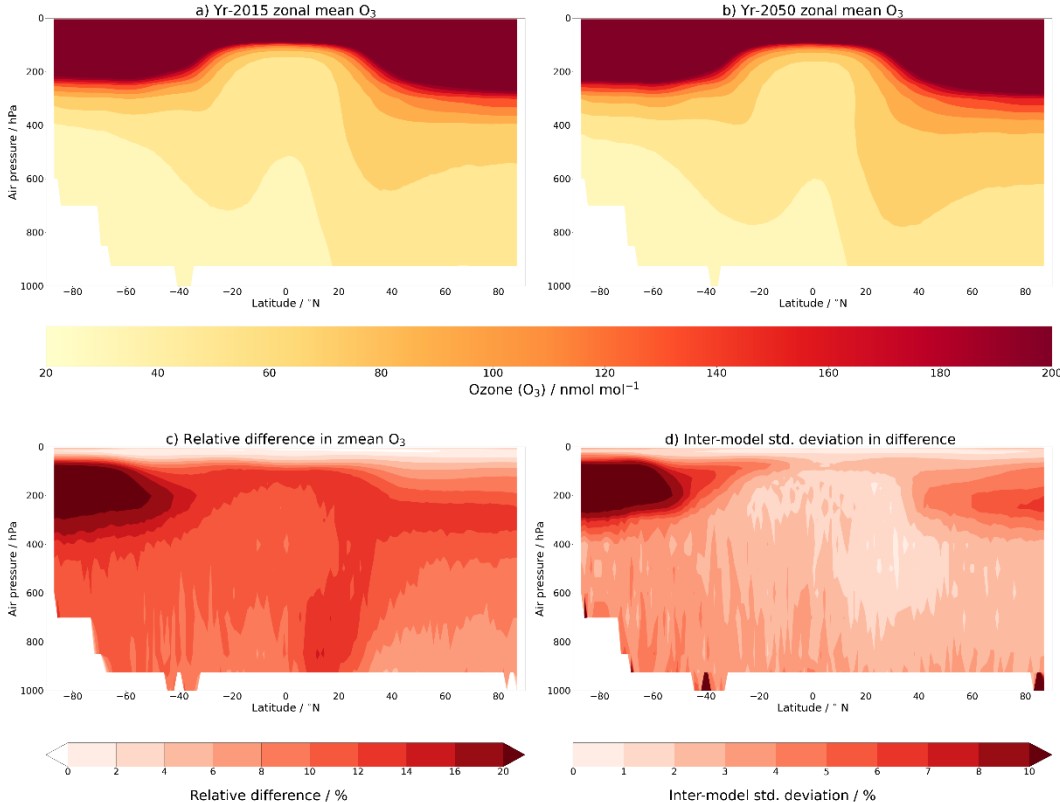


**Figure 5: Multi-model zonal mean climatologies of a) present-day (year 2015) and b) future (year 2050) ozone distributions, c) the multi-model mean relative difference between the climatologies (year-2050 minus year-2015), and d) the inter-model standard deviation about the multi-model mean relative difference. Units of ozone are in nmol mol-1 in panels a) and b). Models included in the multi-model means are CESM2, EMAC, GEOS-Chem, GFDL-ESM4, GISS-E2.1_FR, NorESM2, and UKESM1-0-LL.**

**3.2 Ozone evaluation**

As a result of the bespoke changes made to the different models to meet the requirements of the experimental protocol, it is important to assess any potential impact they may have had on model performance. As result, modelled ozone is benchmarked against CMIP6 historical and future SSP3-7.0 simulations and observations, where available. Figure 6 shows a comparison of modelled TCO between 60 °S and 60 °N from the *pdClim-control* and *pdClim2050ssp370-radO3* simulations against the
CMIP6 ensemble and observations (Bodeker Scientific, 2024). It shows that the present-day (year 2015) ensemble mean TCO of 294.5±9.5 DU is in excellent agreement with that from CMIP6 (297.5±12.7 DU; Keeble et al., 2021). However, like the CMIP6 ensemble mean, the TOAR-RF ensemble mean is systematically biased high relative to observations by approximately 10 DU. In 2050, the TOAR-RF area-weighted TCO ensemble mean is projected to increase to 305.2±12.3 DU following



SSP3-7.0 and is again consistent with that from the CMIP6 ensemble (308.5±11.8 DU; Keeble et al., 2021). This suggests that

the bespoke changes did not have a negative impact on modelled TCO performance relative to CMIP6.



**Figure 6: Comparison of the climatological total column ozone (TCO) averaged over 60°S-60°N and its variability expressed as ±1 standard deviation from the timeslice simulations for the years 2015 and 2050 from all of the individual models (CESM2 in light green; EMAC in orange; GEOS-Chem in blue; GFDL-ESM4 in red; GISS-E2.1_FR in light blue, the GISS-E2.1_nudged model in magenta; NorESM2 in purple and UKESM1-0-LL in dark green) in comparison with the CMIP6 multi-model mean (black solid line) and ±1 standard deviation (grey shading) from coupled transient historical and future simulations using SSP3-7.0, including historical observations from NIWA-BS (black triangles). Individual model results are slightly offset from 2015 and 2050 for greater clarity. The multi-model mean (black diamond), and the inter-model spread expressed as ±1 standard deviation is based on the models: CESM2, EMAC, GEOS-Chem, GFDL-ESM4, GISS-E2.1_FR, NorESM2, and UKESM1-0-LL. Figure adapted from Keeble et al. (2021).**

In terms of individual model performance, present-day TCO from NorESM2 (282.7±2.8 DU) and GFDL-ESM4 (282.3±2.4

DU) sits just outside the lower edge of the ±1 standard deviation CMIP6 multi-model envelope, indicating that both models

are in excellent agreement with the observations (283.5±1.1 DU from the 2010-2014 time period; Bodeker Scientific, 2024).

At the other end of the TOAR-RF ensemble, GEOS-Chem has the highest present-day TCO values of 308.9±4.6 DU, with a



systematic bias of more than 25 DU with respect to the observations. However, it is still within the ±1 standard deviation of the CMIP6 multi-model ensemble. It is also worth noting that modelled TCO from StratTropv2.0 in UKESM1-0-LL is well within the inter-model spread of the CMIP6 and the TOAR-RF ensembles, whereas TCO from StratTropv1.0 was biased high relative to both CMIP6 and observations (Keeble et al., 2021).


## 3.3 Online radiative forcing

Here we intercompare the ozone forcing metrics calculated online by the global models. Figure 7 shows the global mean online calculated IRF, SARF and ERF corresponding to changes between the simulations *pdClim-2050ssp370-radO3* and *pdClim-control*. Results of the GISS-E2.1_nudged model are excluded in Figure 7 and in the following description as in Sect. 3.1.


The multi-model mean ERF and IRF are 0.27 ± 0.09 Wm-2 (6 models) and 0.29 ± 0.10 Wm-2 (5 models), respectively (see Tables S1 - S3). All models, except for GISS-E2.1_FR, indicate a significant positive ERF (see Table S3 in Supplement). Similarly, all models indicate a significant positive IRF (see Table S1). Individual models do not agree whether rapid adjustments enhance or reduce the ERF, as for some models ERF is greater than IRF (GISS-E2.1_FR, UKEMS1-0-LL), whereas for others ERF is smaller than IRF (EMAC, GFDL-ESM4). The partitioning between LW and SW radiative forcing changes from IRF to ERF. The multi-model mean SW all-sky ERF is enhanced compared to SW IRF, whereas the LW all-sky ERF is reduced compared to LW IRF. Also, the individual models agree that the SW all-sky ERF is reduced in comparison to SW all-sky IRF.


SARF was calculated as an online diagnostic only by EMAC indicating a SARF of 0.19 ± 0.00 Wm$^{-2}$ (see Table S2). The SARF is reduced by 0.08 Wm$^{-2}$ compared to the IRF.


The spatial distribution of online calculated all-sky and clear-sky ERF, IRF and SARF from the EMAC model simulation is shown in Fig. S6. The EMAC model was used as it is the only model that provides all the online diagnostics, and we wanted to ensure a fair comparison of the spatial distributions across the metrics by using the same model throughout. Except for the all-sky ERF, all other forcing diagnostics show positive flux changes throughout the globe in EMAC. For all-sky ERF, the fluxes are considerably noisy as this diagnostic is calculated from two simulations (*EMAC-pdClim-2050ssp370-radO3* minus *EMAC-pdClim-control*), which feature different cloud fields (see Sect. 3.5). The largest values of ERF and IRF (both clear-sky and all-sky) can be found in the SH polar region (Fig. S6a) related to changing ODS abundances in the two simulations (see Sect. 3.6 for details). For SARF the high southern latitude forcing is much reduced by the stratospheric temperature adjustment and the largest flux changes are found in the NH at around 25°N. Local maxima of IRF/SARF (both clear-sky and all-sky) and clear-sky ERF can be found over northern Africa, the Arabian Peninsula and the northern part of the Indian subcontinent resembling the largest changes in tropospheric column ozone found in these regions in Fig. 4c and Fig. S4.





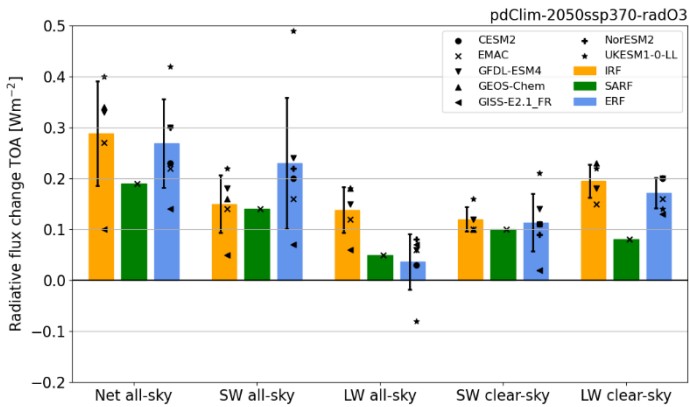

**Figure 7: Online calculated IRF, SARF and ERF corresponding to the standard perturbation *pdClim-2050ssp370-radO3* minus *pdClim-control*. The multi-model mean represented by the colored bars accounts for all models in the TOAR-RF ensemble, which provided the respective diagnostic, which are different for each diagnostic. The multi-model spread is given by the inter-model standard deviation. The markers represent individual model estimates.**

### 3.4 Offline radiative forcing

The SARF at TOA from the ozone changes between *pdClim-2050ssp370-radO3* and *pdClim-control* are calculated offline using the radiative kernel (see section 2.4). The results are shown in Fig. 8 where contributions from total ozone radiative forcing are shown as coloured bars, and the tropospheric forcing is indicated by hatching. The SARF for total ozone range from 0.13 W m$^{-2}$ to 0.31 W m$^{-2}$ (Table S4) with 57 to 93% contribution from ozone in the troposphere ranging from 0.12 W m$^{-2}$ to 0.19 W m$^{-2}$ (Table S5). The multi-model mean SARF is $0.24 \pm 0.057$ W m$^{-2}$ for total ozone and $0.15 \pm 0.019$ W m$^{-2}$ for tropospheric ozone.

The TOA SARF is dependent on latitude and altitude of the ozone change (Fig. S2f) and these dependencies come mainly from the LW (Fig. S2e). For the SW radiative forcing, there are less dependencies on latitude and longitude (Fig. S2b) and the SW RF, including the split between tropospheric and stratospheric contributions, reflect changes in the ozone burden (Fig. S3, Fig. S4). For the LW adjusted radiative forcing, the magnitude is dependent on where ozone changes occur. The increase in stratospheric ozone in the models contribute to a negative LW adjusted radiative forcing, so the total LW adjusted radiative forcing is less than the tropospheric contribution except for NorESM2 (Fig. 8).




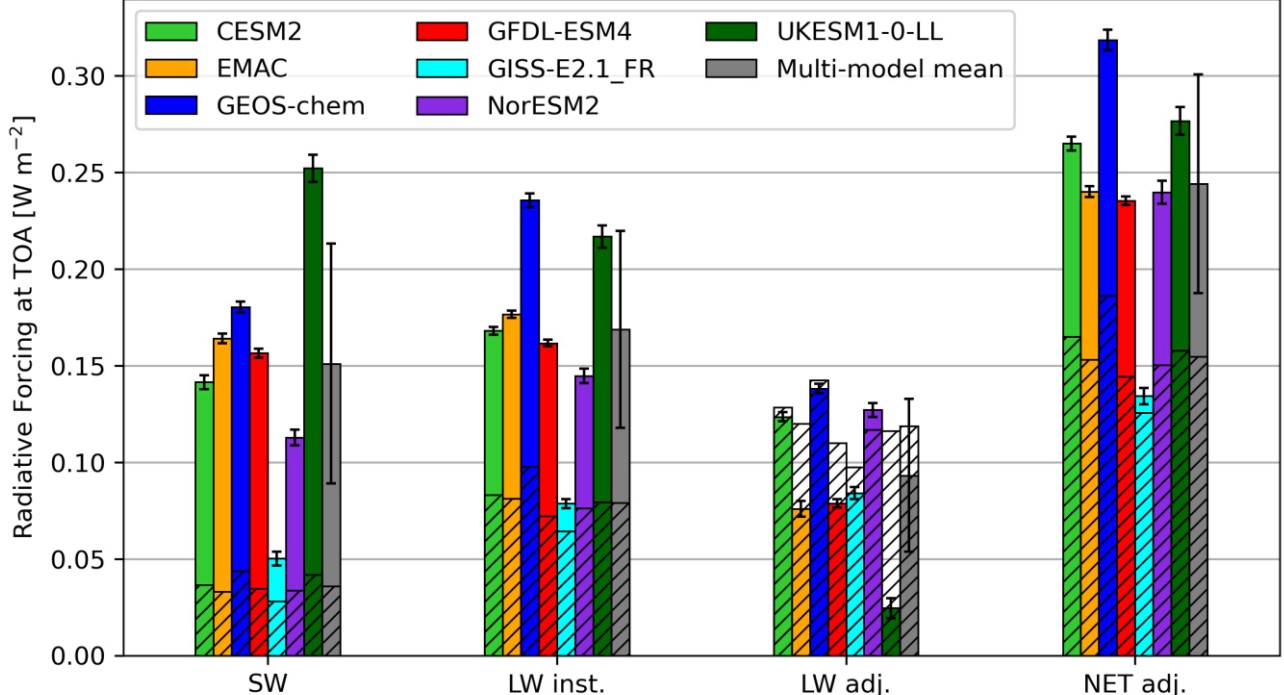


**Figure 8: The radiative forcing at TOA calculated offline using the radiative kernel. The results are separated for short wave (SW), long wave instantaneous (LW inst.), long wave including stratospheric temperature adjustments (LW adj.) and the net SARF (NET adj.) The total ozone forcing for the individual models are shown as a coloured bar and the corresponding standard error indicated by the error bar. The multi-model means are shown as grey bars to the right, where the error bars indicate the standard deviation**
**of the multi-model spread. The hatched parts of the bars are the contribution from the tropospheric ozone changes where the tropopause is defined by 150 nmol mol⁻¹ ozone mole fraction in the *pdClim-control* simulations.**

### 3.5 Cloud changes

The ERF from the ozone perturbation includes adjustments due to changes in clouds. To study this in more detail, we focus
on global vertical profiles of ozone, temperature and cloud fraction, cloud cover and TOA radiative fluxes. We exclude from this analysis GISS-E2.1_nudged as it shows rather different ozone and cloud responses and GEOS-Chem which as a CTM does not allow for cloud changes. The top row of Fig. 9 shows the difference in the global mean profile of ozone, temperature and cloud fraction between the *pdClim-2050ssp370-radO3* and *pdClim-control* simulations. The difference in ozone (shown for 6 of the models in Fig. 9a,b) is mostly positive and strongest above 200 hPa (when expressed in mole fraction). All six
models shown here are in reasonable agreement, except for GISS-E2.1_FR showing only a weak stratospheric signal. In the troposphere, all models indicate an increase in ozone (which is more visible when expressed in partial pressure in Fig. 9b), with GISS-E2.1_FR showing a weaker response than the other 5 models. The difference in temperature between both simulations (shown in Fig. 9c) is most pronounced and positive between 300 and 50 hPa, and negative above 50 to 20 hPa.





The vertical profile of the cloud fraction (shown in Fig. 9d) shows in general a decrease, and this signature is strongest between

400 and 100 hPa. Some models show a secondary peak in the reduction of the cloud fraction at around 800 hPa (CESM2, NorESM2 and UKESM1-0-LL), although other models do not. EMAC (orange line) shows a relatively weak difference for temperature and cloud fraction, and this might be caused by the nudging in EMAC.

The bottom row in Fig. 9 (panels e, f, g, h), shows results for the sensitivity experiment *pdClim-2050ssp370fODS-radO3* (see

Section 3.6) using dotted lines. For the three models that performed the *pdClim-2050ssp370fODS-radO3* experiment (EMAC, GFDL-ESM4 and UKESM1-0-LL), we find a weaker but still coherent response in the vertical profiles of these variables.

The impact seen in the vertical profile of the cloud fraction can also be observed in the change of the total cloud cover, which is an integrated value over the whole atmospheric column. Table 3 gives the global mean cloud cover in the reference

simulation *pdClim-control* (in the second column) and the difference between *pdClim-2050ssp370-radO3* and *pdClim-control* (third column) for the different models. UKESM1-0-LL shows the strongest reduction in cloud cover, followed by slightly weaker responses in NorESM2, CESM2 and GFDL-ESM4, and a relatively small response in EMAC. The cloud cover response for GEOS-Chem (zero) and GISS-E2.1_nudged (positive) are also listed in Table 3 but are not further used in the analysis. Table 3 also shows the contribution of clouds to the TOA SW and LW ERFs. These values have been obtained as

the difference between the all-sky (AS) and clear-sky (CS) TOA ERFs. Figure 10 shows this cloud contribution to the TOA ERFs as a function of the change in cloud cover. Especially for the SW ERF (Fig. 10a) we find a strong relationship, but also for the LW ERF (Fig. 10b) the relation is still present, although somewhat weaker. Figure 10c shows the net (SW+LW) contribution of clouds to the TOA ERF. It is the difference between two relatively large numbers and shows little correlation with the cloud cover change. Figure 10 contains both the impacts of the standard experiment *pdClim-2050ssp370-radO3*

(circles) and of the sensitivity experiment *pdClim-2050ssp370fODS-radO3* (triangles). The *pdClim-2050ssp370fODS-radO3* experiment is more deeply discussed in the next Section 3.6.



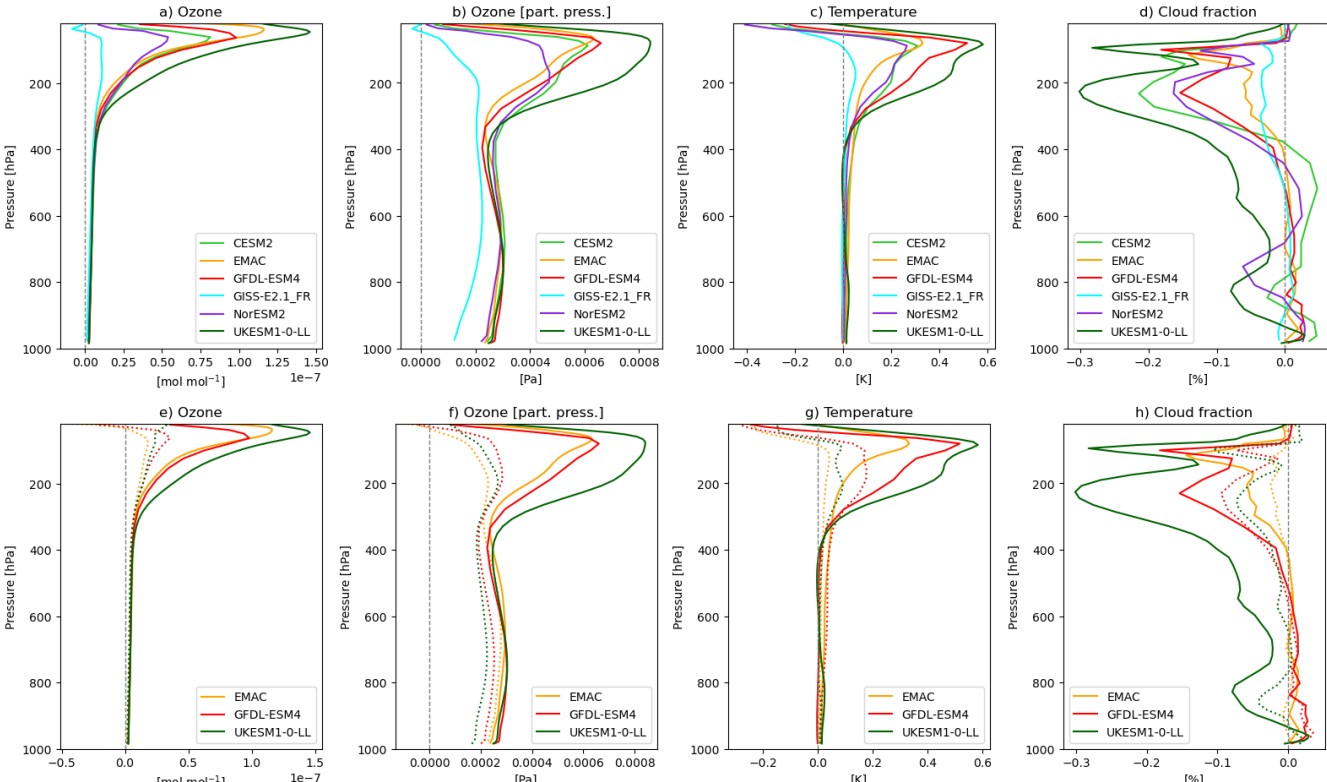

**Fig 9: Global mean profile of difference in (a, e) ozone mole fraction, (b, f) ozone partial pressure (Pa), (c, g) temperature (K) and**
**(d, h) cloud cover (%) between the 2050 and 2015 (*pdClim-control*) state. The top row panels show the results from the standard**
**experiment *pdClim-ssp2050ssp370-radO3* minus *pdClim-control*, the bottom panels show for the models that ran the *fODS-***
***perturbation* experiment (EMAC, GFDL-ESM4, UKESM1-0-LL) the results from the standard (*pdClim-ssp2050ssp370-radO3* minus**
***pdClim-control*) experiment using solid lines (which are repeated from panels a-d) and fODS (*pdClim-ssp2050ssp370fODS-radO3***
**minus *pdClim-control*) experiments using dotted lines.**




| Model | Cloud Cover | | (All-sky minus clear-sky) ERF | | |
|---|---|---|---|---|---|
| | Reference [%] | Absolute Change [%] | SW ERF [W m⁻²] | LW ERF [W m⁻²] | Net ERF [W m⁻²] |
| CESM2 | 69.12±0.02 | -0.08±0.04 | 0.10±0.06 | -0.17±0.03 | -0.08±0.05 |
| EMAC | 62.37±0.11 | -0.03±0.01 | 0.05±0.01 | -0.09±0.03 | -0.04±0.01 |
| GEOS-Chem | 60.69±0.09 | 0±0 | 0.04±0.003 | -0.08±0.005 | -0.04±0.01 |
| GFDL-ESM4 | 66.02±0.02 | -0.08±0.03 | 0.10±0.03 | -0.14±0.02 | -0.04±0.03 |
| GISS-E2.1_FR | 60.03±0.04 | -0.04±0.04 | 0.05±0.03 | -0.06±0.02 | -0.01±0.03 |
| GISS-E2.1_nudged | 61.30±0.003 | 0.124±0.003 | | | |
| NorESM2 | 61.96±0.03 | -0.10±0.04 | 0.10±0.06 | -0.10±0.02 | -0.004±0.07 |
| UKESM1-0-LL | 69.14±0.03 | -0.20±0.03 | 0.28±0.03 | -0.22±0.02 | 0.05±0.02 |
| Multi-model mean | 64.8±3.6 | -0.09±0.06 | 0.11±0.08 | -0.13±0.05 | -0.02±0.04 |

**Table 3: Global mean cloud cover in reference simulation (pdClim-control), absolute difference in cloud cover between both simulations (*pdClim-2050ssp370-radO3* minus *pdClim-control*), and contribution of clouds to the SW, and LW ERF. The uncertainty**
**for the individual models is the error on the mean. The multi-model mean is calculated using the 6 models CESM2 (average of the two members), EMAC, GFDL-ESM4, GISS-E2.1_FR, NorESM2 and UKESM1-0-LL. The uncertainty on the multi-model mean is the standard deviation between the models.**

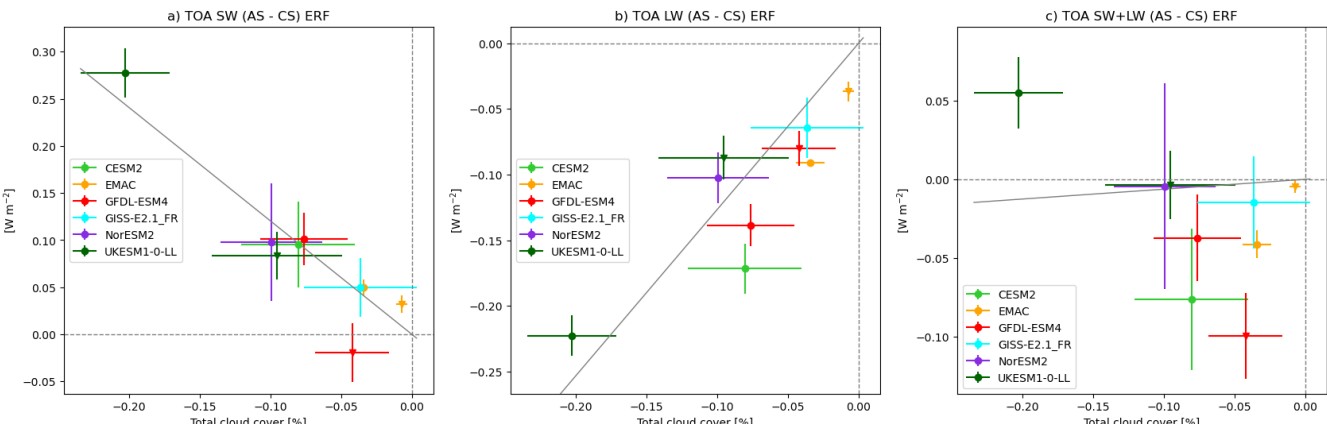


**Fig 10: TOA all-sky minus clear-sky short-wave (a), long-wave (b) and short- plus long-wave (c) ERF (in W m⁻²) as a function of the total cloud cover change (in %). Results from the standard experiment *pdClim-2050ssp370-radO3* are represented by circles, and results from the sensitivity experiment *pdClim-2050ssp370fODS-radO3* are represented by triangles for the three models that did those experiments (EMAC, GFDL-ESM4 and UKESM1-0-LL). The error bars indicate the error on the mean, and the line is a best**
**least-square fit (going through the origin).**





### 3.6 Sensitivity of Forcing to Tropospheric Ozone Precursors

In this section, the sensitivity of the year-2050 ozone response and the corresponding forcings will be examined, making use of the *pdClim-2050ssp370fODS-radO3* (*fODS-perturbation*) simulation (Table 1), which was carried out by three of the models: EMAC, GFDL-ESM4, and UKESM1-0-LL. The global mean nitrous oxide ($N_2O$) concentration increases from a present-day value of 328.18 to 361.78 nmol mol$^{-1}$ in the *pdClim-2050ssp370fODS-radO3* (*fODS-perturbation*) simulation and may be important for ozone depletion. However, our focus here is on *pdClim-2050ssp370fODS-radO3* minus *pdClim-control* tropospheric ozone differences, which will be predominantly driven by increases in tropospheric ozone precursors in 2050 following SSP3-7.0. Figure 11 shows the change in TCO, TrCO, and zonal mean ozone in 2050 relative to the present day from the smaller 3-member ensemble. In 2015, the multi-model global mean TCO of 297.4 DU is within 1 DU of the full 7-member TOAR-RF ensemble mean shown in Fig. 3 (298.3 ± 8.3 DU), indicating that the 3-member ensemble is reasonably representative of the larger 7-member ensemble. Global mean TCO increases by 4.8 DU to 302.1 DU in 2050 in the *fODS-perturbation* simulation, with the largest increases occurring over the middle East, India, southeast Asia and the high latitudes in both hemispheres (Fig. 11a). The individual global mean TCO increases range from 3.8 DU in the EMAC model to 5.7 DU in UKESM1-0-LL, with the GFDL-ESM4 ozone increase (4.9 DU) close to the 3-member ensemble mean. Thus, the TCO increase in the *fODS-perturbation* explains over 30 % of the TCO increase in *pdClim-2050ssp370-radO3* (15.1 DU; Fig. 11a).

Turning to TrCO, the present-day global mean value of 36.7 DU is again consistent with the 7-member ensemble (36.2 ± 1.1 DU; Fig. 4). In the *fODS-perturbation* simulation, year-2050 TrCO increases by 3.3 DU to 39.9 DU. A comparison of TrCO changes in *fODS-perturbation* (3.3 DU; Fig. 11d) with those from *pdClim-2050ssp370-radO3* (4.3 DU; Fig. 11cd) indicates that over 75 % of the increase in TrCO is driven by tropospheric ozone precursors. This is evident over the middle East, India, and southeast Asia including its outflow regions, where increases in TrCO of 4-8 DU occur. From a zonal mean perspective, Fig. 11f shows that the largest relative changes in ozone in the *fODS-perturbation* simulation occur in the southern high latitude upper troposphere and in the northern hemisphere troposphere. This latter region of large relative increase extends from the surface into the upper troposphere (100-300 hPa) in the tropics and extra-tropics, where the radiative efficiency for ozone forcing is highest (Fig. 1).



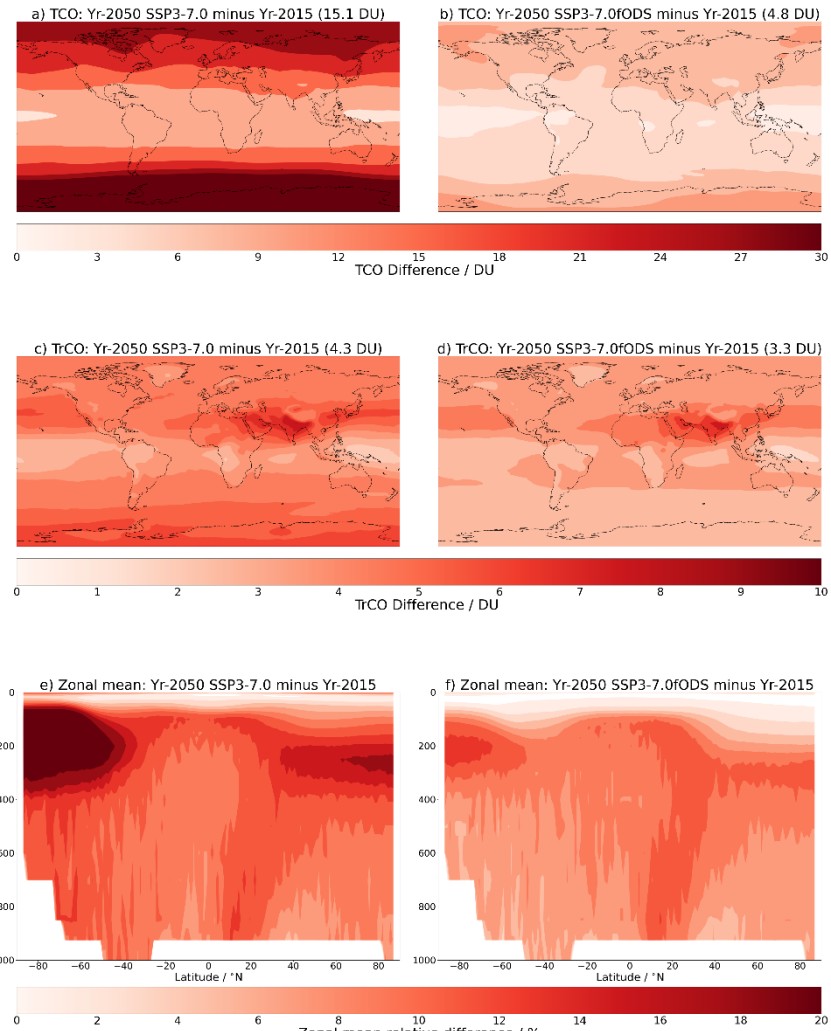

**Figure 11: Multi-model mean differences in total column ozone (TCO; panels a and b), tropospheric column ozone (TrCO; panels c and d), and zonal mean ozone (panels e and f) in 2050 relative to the present day from *pdClim-2050ssp370-radO3* (left column) and *pdClim-2050ssp370fODS-radO3* (fODS; right column). Panels a), c), and e) are the same as Figs. 3c), 4c), and 5c), respectively, but only those models that performed the fODS-perturbation *pdClim-2050ssp370fODS-radO3* are shown here as it provides a fairer comparison to panels b), d), and f). Units of TCO and TrCO differences are in Dobson Units (DU), while the zonal mean differences are in %. Models included in the multi-model means are EMAC, GFDL-ESM4, and UKESM1-0-LL.**

The RF estimates presented in Sect. 3.3 and 3.4 take account of changes in stratospheric ozone due to the expected reduction in ODSs (Daniel et al., 2022). Here, the global mean all-sky and clear-sky ERF is reduced in the *fODS-perturbation* compared to the standard perturbation for all models (see Figure 12). The global multi-model mean all-sky ERF of the three models that performed *pdClim-2050ssp370fODS-radO3* decreases from $0.31 \pm 0.08$ Wm$^{-2}$ in the standard perturbation *pdClim-2050ssp370-radO3* to $0.15 \pm 0.03$ Wm$^{-2}$ in *pdClim-2050ssp370fODS-radO3*. The all-sky ERF is reduced by around 60% for



GFDL-ESM4 and UKESM1-0-LL, and by around 23% for EMAC. In the clear-sky, both LW and SW ERF are reduced, with changes in clear-sky ERF being larger in the SW for all models.


**Figure 12: Online calculated IRF, SARF and ERF corresponding to (a) the standard perturbation _pdClim-2050ssp370-radO3_ and (b) the fODS-perturbation _pdClim-2050ssp370fODS-radO3_ for models that performed the fODS-perturbation _pdClim-2050ssp370fODS-radO3_. Panel (a) is the _same_ as Figure 7, but only those models that performed the fODS-perturbation _pdClim-2050ssp370fODS-radO3_ are shown here as it provides a fairer comparison to panel (b).**

Global mean all-sky and clear-sky IRF is reduced in the _fODS-perturbation_ compared to the standard perturbation simulation for all models. This is due to both reduced IRF in the SW and in the LW. As can be expected, the increased stratospheric ozone
because of the reduction in ODSs in the standard perturbation _pdClim-2050ssp370-radO3_ leads to additional absorption of SW radiation compared to the _fODS-perturbation_. In the LW, the trapping of outgoing LW radiation by the additional stratospheric ozone seems to dominate, because LW IRF is larger in the standard perturbation compared to the _fODS-perturbation_. The global multi-model mean all-sky IRF of the three models that performed _pdClim-2050ssp370fODS-radO3_ decreases from $0.33 \pm 0.05$ Wm$^{-2}$ in the standard perturbation _pdClim-2050ssp370-radO3_ to $0.13 \pm 0.04$ Wm$^{-2}$ in _pdClim-_
_2050ssp370fODS-radO3._,

EMAC is the only model that provides online calculated SARF for _pdClim-2050ssp370fODS-radO3_. The corresponding global mean all-sky SARF is estimated at $0.14 \pm 0.00$ Wm$^{-2}$ (see Table S6). As discussed in section 3.3, the stratospheric temperature adjustment is negative (IRF>SARF) in the standard perturbation _pdClim-2050ssp370-radO3_ for EMAC. This is consistent
with the temperature in the stratosphere increasing due to increased stratospheric ozone levels (see Figure 9g), which leads to a reduced LW forcing for SARF compared to IRF. In the _fODS-perturbation_, however, SARF is enhanced by 0.06 Wm$^{-2}$ compared to IRF due to the decreased stratospheric temperatures (Fig. 9g).



From the offline kernel calculation, the total ozone SARF for the *fODS-perturbation* are 0.18, 0.17 and 0.16 W m$^{-2}$ for EMAC,
GFDL-ESM4 and UKESM1-0-LL respectively (Table S7). For EMAC and GFDL-ESM4, this is 75% and 72% of the total
ozone SARF calculated from *pdClim-2050ssp370-radO3* and *pdClim-control*, while for UKESM1-0-LL the contribution is
only 57%. Tropospheric ozone, where the tropopause is defined by the monthly ozone mole fraction in the control simulation,
contributes between 71% and 77% of the total ozone SARF in the *fODS-perturbation* for all three models. The tropospheric
ozone SARF is presented in Table S8.


We have also analysed the latitudinal distribution of radiative flux changes in the EMAC *fODS-perturbation* simulation (Fig.
S7b). We focus here on EMAC as it is the only model that provides all online diagnostics. Apart from the noisy behaviour of
the all-sky ERF, ERF values are generally larger than SARF values, which are larger than IRF values at almost all latitudes.
This holds true for both clear-sky and all-sky diagnostics. Comparing the latitudinal distribution of the forcings in the standard
perturbation and the *fODS-perturbation*, the largest changes can be found in the polar regions, in particular over the Antarctic,
which highlights the impact of changing ODS abundances on the RF diagnostics.

An additional effect that we cannot isolate with the *fODS-perturbation* simulation is the impact of increasing $N_2O$ emissions
in the SSP3-7.0 scenario on ozone. In the stratosphere, $NO_x$ is produced from $N_2O$ and hence $N_2O$ is foreseen to be an important
factor with respect to ozone depletion in the 21$^{st}$ century (Ravishankara et al., 2009; Revell et al., 2012). Prescribing increasing
$N_2O$ surface mixing ratios leads to an ozone reduction in the middle and upper stratosphere, which would counteract the
increase of stratospheric ozone related to the expected decrease in ODSs until 2050 (Kirner et al., 2015; results based on a
different emission pathway), despite ozone loss due to $NO_x$ being less efficient when $CO_2$ and $CH_4$ concentrations increase
(Revell et al., 2015).


## 4. Discussion

The future increases in ozone precursors and decreases in ozone depleting substances in the SSP3-7.0 scenario lead to an
increase in global mean TCO by 12.2 DU between 2015 and 2050 in the multi-model mean with the 7 individual models
included in the ensemble mean showing increases of 4.3 to 19.6 DU. From a multi-model mean perspective, about 4.3 DU
(around 39%) of the global mean TCO changes are attributable to the tropospheric ozone changes, while the rest is due to
changes in stratospheric ozone abundances. In a sensitivity simulation in which ODSs in 2050 were kept at year-2015 levels
(fODS), the importance of ozone precursor changes was assessed by a subset of models. In the respective multi-model mean,
the global mean TCO increases by 4.8 DU, with individual model increases ranging from 3.8 DU in EMAC to 5.7 DU in
UKESM1-0-LL (see Sect. 3.6).






## 4.1 Radiative efficiencies

Based on the model dependent changes in ozone with respect to the standard perturbation simulation, an offline radiative kernel leads to a total SARF of 0.13 to 0.31 $Wm^{-2}$. With 34% to 43% of this forcing coming from changes in ozone above the tropopause (defined as ozone mole fraction values of 150 nmol $mol^{-1}$; see Sect. 3.4). When scaling the kernel SARF by the
total column ozone change (see Fig. 13), the radiative efficiency varies from 0.014 to 0.030 $Wm^{-2} DU^{-1}$. For models that show lower increases in stratospheric ozone, their ozone radiative efficiency is towards the higher end of this range, whereas for models showing higher increases in stratospheric ozone, the efficiency is towards the lower end. This is due to the kernel having the highest efficiency in the upper troposphere and lower stratosphere (Fig. 1). When using only the tropospheric column, there is much more agreement on the radiative efficiency with a range of 0.039 to 0.044 $Wm^{-2} DU^{-1}$, which is in good
agreement with previous findings of a tropospheric ozone radiative efficiency of 0.042 $Wm^{-2} DU^{-1}$ in Stevenson et al. (2013). For the *fODS-perturbation*, the radiative efficiencies for the total column and tropospheric column are in closer agreement reflecting the smaller effects on stratospheric ozone.

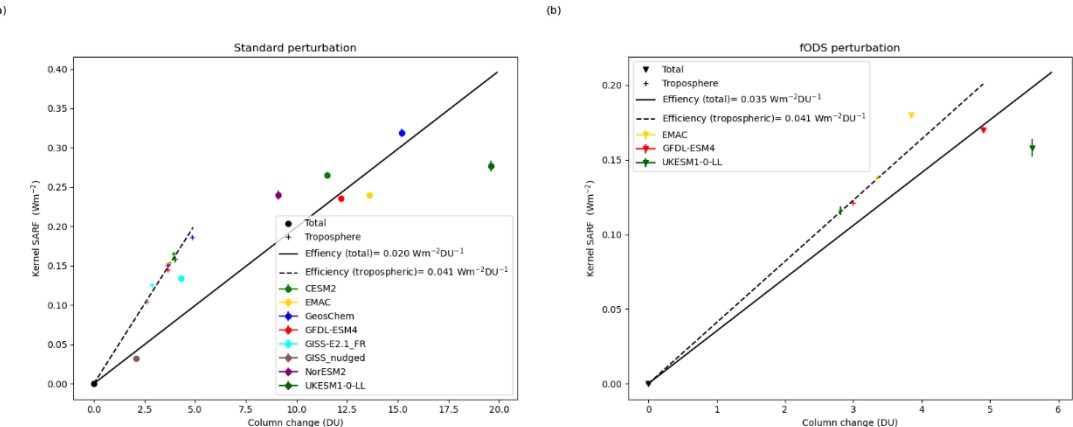

**Figure 13: Kernel SARF vs column ozone change for (a) standard perturbation, (b) *fODS-perturbation*. Both tropospheric-only and**
**total column changes and SARF are shown.**

## 4.2 Instantaneous radiative forcing

The ozone IRFs are calculated both by the kernel method (Sect. 3.3) and by online double calls to the radiation scheme (Sect. 3.3). There is strong correlation between the two methods (see Fig. 14), however the kernel-calculated IRFs are larger than those calculated online, by 26% in the LW.





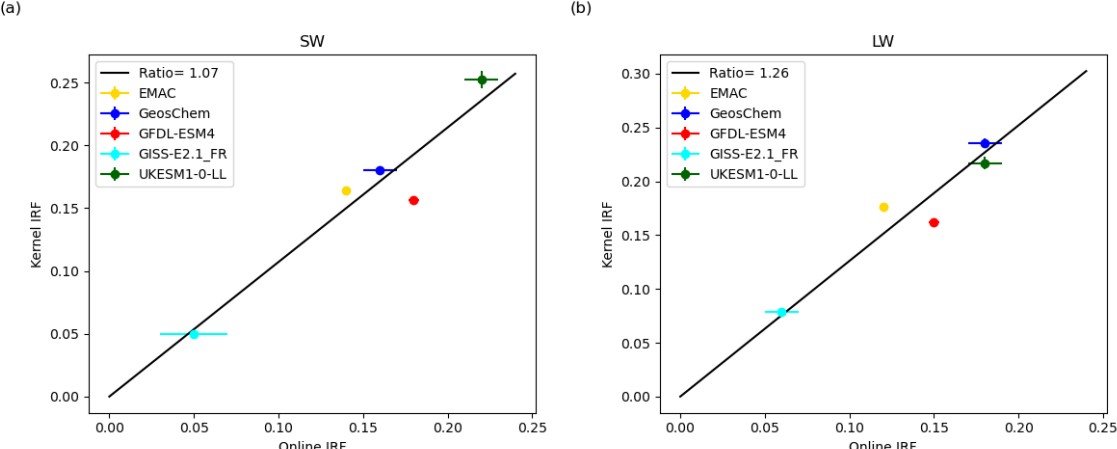


**Figure 14: Comparison of the online (double call) and kernel methods for calculating the instantaneous radiative forcing (IRF) for (a) SW and (b) LW.**

## 4.3 Non-cloud adjustments

For changes in tropospheric ozone (Table S5), as a result of stratospheric-temperature adjustments, the SARF calculated by
the kernel is between +31% to +38% greater than the kernel-calculated instantaneous forcing. Since the adjustments are pre-
calculated in the kernels, the variation is due to slight variations in the vertical distribution of the ozone changes in the different
models. The adjustments are a lot smaller than in Shine et al. (2022) which found around +80% adjustment for a pre-industrial
to present day tropospheric ozone change (Checa-Garcia et al., 2018). The adjustments to changes in stratospheric ozone are
a lot more variable, from -40% to -65% reflecting the wider variability in the vertical distributions of the ozone changes in the
stratosphere.  The adjustments in Shine et al. (2022) were larger, -80% of the IRF for stratospheric ozone changes, but these
were dominated by the depletion caused by ODSs rather than in this study where the changes in ozone precursors make
substantial contributions in the stratosphere.

By excluding cloud effects, the clear sky ERFs should compare to the kernel SARFs if the tropospheric non-cloud adjustments
are small. The modelled clear-sky SW ERF correlates well with the SARF kernel calculations (see Fig. 15) although the ERF
calculation is consistently higher, the ratio depending on whether ODS changes are included (standard) or not (fODS). The
clear-sky SW ERF is also higher than the online IRF (Tables S1 and S3). This implies a positive clear-sky adjustment. The
largest clear-sky SW adjustment would be expected to come from the surface albedo change such as snow cover, which is
expected to be positive (Smith et al., 2020). We do not have a clear-sky LW SARF kernel, so we scale the all-sky LW SARF
kernel by the ratio of  the clear-sky to all-sky LW IRF kernel. In the LW there is a positive correlation between clear-sky ERF
and SARF, with closer correlation when ODS changes are excluded. In this case (fODS), the ERF results are lower than the
SARF kernel  which is in line with the kernel IRF being larger than the online calculations in the models. Any negative
adjustments due to the increases in tropospheric temperature seen in Fig. 9g are likely to be largely offset by positive





adjustments due to increases in water vapour (Smith et al., 2020). For the standard perturbation, the larger LW clear-sky ERF

than kernel SARF implies positive adjustments that are likely to be due to increases in water vapour. These conclusions of

little clear-sky LW adjustment in the fODS perturbation and a positive adjustment in the standard perturbation are also borne

out by comparing the SARF and ERF for the only model (EMAC) that can calculate an online SARF (tables S1, S2, S6).

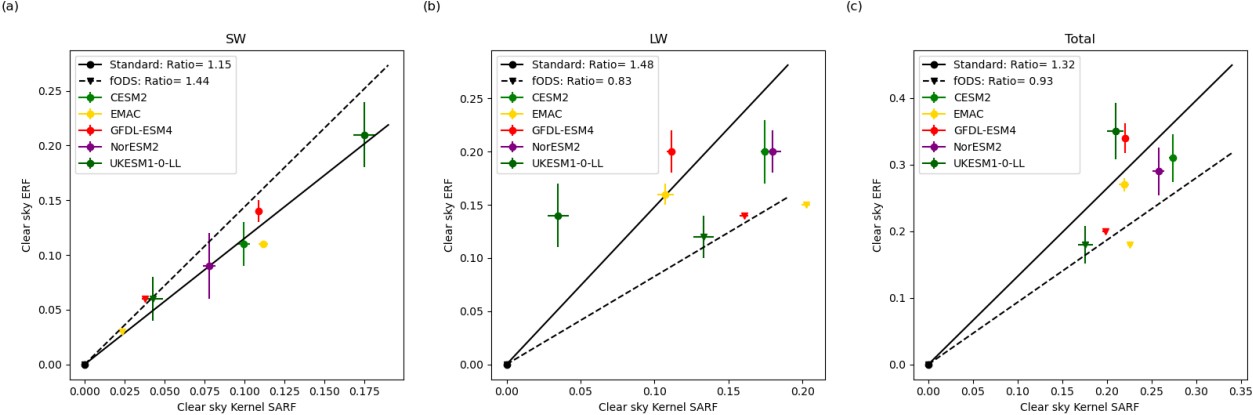

**Figure 15: Comparison of clear sky ERF and SARF from the kernel calculations. For (a) SW, (b) LW and (c) total forcing. Both**
**standard and fODS perturbations are plotted.**

### 4.4 Cloud adjustments

There are large adjustment terms found to be due to changes in cloud fraction (Sect. 4.6, Table 3). Note that the difference

between all-sky and clear-sky ERF isn't solely due to cloud adjustments but includes a contribution from cloud masking of

the ozone forcing. The models agree on a reduction in upper tropospheric cloud fraction, and (apart from UKESM1-0-LL) an

increase in the mid-troposphere. In the lower troposphere there is less agreement, with some models showing a decrease in

cloud fraction around 800 hPa and an increase just above the surface. This is generally in agreement with a study by MacIntosh

et al. (2016) which found stratospheric ozone depletion increased high cloud, and upper tropospheric increases decreased high

and medium level cloud but increased low cloud. There are strong correlations between the changes in cloud fraction and the

cloud adjustments in the SW and a weaker correlation in the LW. Reduced cloud cover gives a positive SW adjustment due to

decreased planetary albedo, and a negative LW adjustment as reductions in high cloud increase the overall emitting temperature

of the atmosphere. These two largely cancel, with the larger LW adjustment leading to an overall negative cloud adjustment.

The exceptions are UKESM1-0-LL (standard perturbation) which has a strong positive SW adjustment, and NorESM2

(standard perturbation) which has a weaker LW adjustment. The cloud changes and associated SW and LW adjustment in the

*fODS-perturbation* are smaller, highlighting the stronger effect of stratospheric ozone changes on clouds. Because the overall

adjustment is the compensation of two opposing terms, there is no correlation with the change in cloud fraction and no

systematic difference between the standard and fODS perturbations. Nevertheless, the cloud adjustment (-0.1 to +0.05 Wm$^{-2}$)

is a significant fraction of the total ERF (0.27±0.04 Wm$^{-2}$).



# 5. Conclusions

We have shown that projected increases in tropospheric ozone precursors and decreases in ODSs in the SSP3-7.0 scenario lead to increases in ozone in the troposphere and stratosphere. By restricting the impact of composition changes on the evolution of the physical model, we can isolate the changes solely due to changes in ozone. This contributes an ERF of $0.27\pm0.09$ Wm$^{-2}$ by 2050. A subset of models calculated the ERF excluding ODS changes; for these, the ERF decreased from $0.31 \pm 0.08$ Wm$^{-2}$ in the standard perturbation to $0.15 \pm 0.03$ Wm$^{-2}$ with fixed ODSs. Decreases in ozone between the standard and fODS

perturbations were not confined to the stratosphere but also affected the upper troposphere. This highlights the increase in ozone forcing expected from commitments to reduce ODSs which offsets some of the climate benefits from reducing their direct greenhouse effect.

Few of the models were able to calculate a SARF online, so we used offline kernel calculations to determine the SARF. We

tested the kernel calculations by comparing the IRFs for kernel and online calculations. The kernel calculations give consistently higher IRFs (both LW and SW). The multi-model mean ERF for the standard perturbation is $0.27\pm0.09$ Wm$^{-2}$ compared to the kernel SARF of $0.24 \pm 0.02$ W m$^{-2}$, suggesting an overall positive (non stratospheric temperature) adjustment. This also true for the online ERF vs SARF in the EMAC model. The cloud adjustments are negative (-0.02$\pm$0.04 Wm$^{-2}$) which implies positive non-cloud adjustments. This is supported by the finding of positive adjustments in both SW and LW comparing

clear-sky ERF and SARF. For the *fODS-perturbation* the ERF is $0.15 \pm 0.03$ Wm$^{-2}$, with a kernel SARF of $0.17 \pm 0.01$ Wm$^{-2}$. The overall negative adjustment is due to the more negative cloud adjustment in two of the models (mean of -0.04$\pm$0.04 Wm$^{-2}$) and a less positive non-cloud LW adjustment.

This study shows that care is needed when interpreting or comparing radiative forcing calculations for ozone. There are

discrepancies between the online and offline calculations of the IRF, particularly in the LW. This might indicate differences in the SARF. Ultimately the radiative forcing calculated as ERFs or SARF is similar. This occurs due to a cancellation between a negative cloud adjustment and a positive non-cloud adjustment.

Data Availability

The model data from Phase 6 of the Coupled Model Intercomparison Project (CMIP6) used in Figure 6 of this study are available through the Earth System Grid Federation (ESGF; https://esgf-index1.ceda.ac.uk/projects/cmip6-ceda/, last access: October 2024) (ESGF, 2024), with specific references for each model dataset and the DOIs for each dataset in Table 1 and the reference list of Keeble et al. (2021), respectively. Near-global total column ozone observational data were taken from Version

3.4 of the National Institute of Water and Atmospheric Research – Bodeker Scientific (NIWA-BS) combined TCO database, available from http://www.bodekerscientific.com/data/total-column-ozone (last access: November 2024) (Bodeker Scientific,



2024). The radiative kernel is available from https://github.com/ciceroOslo/Radiative-kernels (last access: November 2024). The data in the figures from this manuscript are available from zenodo with the DOI 10.5281/zenodo.14238129 (O'Connor et al., 2024).

## Acknowledgments

WJC was supported by the UK Natural Environment Research Council Grant "Investigating Halocarbon Impacts on the global environment" (Grant Reference NE/X004198/1). FMO'C was supported by the European Union's Horizon 2020 project ESM2025 (under grant agreement No 101003536) and the Met Office Hadley Centre Climate Programme funded by DSIT, UK. MM was supported by the German Federal Ministry of Education and Research (Funding Nr.: 01LN2207A, IMPAC2T). The work involving the EMAC model used resources of the Deutsches Klimarechenzentrum (DKRZ) granted by its Scientific Steering Committee (WLA) under project ID id0853. Further, datasets provided by MESSy via the DKRZ data pool were used. RB, RBS and GM were supported by the European Union's Horizon 2020 research and innovation program under grant agreement no 820829 (CONSTRAIN project). ØH was supported by the Research Council of Norway (project no. 336227). DO was supported by the European Union's Horizon 2020 project ESM2025 (under grant agreement No 101003536). The authors would also like to thank Bodeker Scientific, funded by the New Zealand Deep South National Science Challenge, for providing the combined NIWA-BS total column ozone database. UI is supported by the European Union's Horizon Europe project CleanCloud (under grant agreement No 101137639).

## Author contributions

WJC and FMO'C  jointly designed the study. Model development and set up was by FMO'C, ØH, CRB, REB, SDE, PJ, MM, MN, DO, RBS,  LS, LWH, VN, GF, UI, EAM, LTM, DS, KT, NLA and JK. FMO'C, REB, PJ, MM, DO, LWH, GF, UI, LTM, DS and KT ran the model simulations. Data analysis and construction of figures was by WJC, FMO'C, REB, ØH, PJ, MM, GM, MN, DO, RBS, LS, VN  and LTM. All authors were involved in drafting and reviewing the manuscript.

## Competing interests

Some authors are members of the editorial board of Atmospheric Chemistry and Physics.

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
