# Peer review of "Climate Forcing due to Future Ozone Changes: An intercomparison of metrics and methods"

_EGUsphere, 2024_

## Community Comment (CC1)

December 20, 2024

Comments by Owen R. Cooper (TOAR Scientific Coordinator of the Community Special Issue) on:

**Climate Forcing due to Future Ozone Changes: An intercomparison of metrics and methods**

Collins, W. J., O'Connor, F. M., Barker, C. R., Byrom, R. E., Eastham, S. D., Hodnebrog, Ø., Jöckel, P., Marais, E. A., Mertens, M., Myhre, G., Nützel, M., Olivié, D., Bieltvedt Skeie, R., Stecher, L., Horowitz, L. W., Naik, V., Faluvegi, G., Im, U., Murray, L. T., Shindell, D., Tsigaridis, K., Abraham, N. L., and Keeble, J.

EGUsphere [preprint], https://doi.org/10.5194/egusphere-2024-3698
Discussion started Dec. 6, 2024
Discussion closes Jan. 17, 2025

This review is by Owen Cooper, TOAR Scientific Coordinator of the TOAR-II Community Special Issue. I, or a member of the TOAR-II Steering Committee, will post comments on all papers submitted to the TOAR-II Community Special Issue, which is an inter-journal special issue accommodating submissions to six Copernicus journals:  ACP (lead journal), AMT, GMD, ESSD, ASCMO and BG. The primary purpose of these reviews is to identify any discrepancies across the TOAR-II submissions, and to allow the author teams time to address the discrepancies.  Additional comments may be included with the reviews. While O. Cooper and members of the TOAR Steering Committee may post open comments on papers submitted to the TOAR-II Community Special Issue, they are not involved with the decision to accept or reject a paper for publication, which is entirely handled by the journal's editorial team.

**Comments regarding TOAR-II guidelines:**

TOAR-II has produced two guidance documents to help authors develop their manuscripts so that results can be consistently compared across the wide range of studies that will be written for the TOAR-II Community Special Issue.  Both guidance documents can be found on the TOAR-II webpage: https://igacproject.org/activities/TOAR/TOAR-II

*The TOAR-II Community Special Issue Guidelines*:  In the spirit of collaboration and to allow TOAR-II findings to be directly comparable across publications, the TOAR-II Steering Committee has issued this set of guidelines regarding style, units, plotting scales, regional and tropospheric column comparisons, and tropopause definitions.

*The TOAR-II Recommendations for Statistical Analyses*:  The aim of this guidance note is to provide recommendations on best statistical practices and to ensure consistent communication of statistical analysis and associated uncertainty across TOAR publications. The scope includes approaches for reporting trends, a discussion of strengths and weaknesses of commonly used techniques, and calibrated language for the communication of uncertainty. Table 3 of the TOAR-II statistical guidelines provides calibrated language for describing trends and uncertainty, similar to the approach of IPCC, which allows trends to be discussed without having to use the problematic expression, "statistically significant".

**General comments:**

The paper begins with an excellent review of ozone radiative forcing, which will be a very helpful reference for the scientific community (especially Figure 1). Thank you.

Abstract.
Here the main findings on ozone RF are reported as:
"We find robust changes in ozone due to future changes in ozone precursors and ODSs. These lead to a positive radiative forcing of 0.27±0.09 Wm-2 ERF, 0.24 ± 0.021 W m-2 offline SARF, 0.29 ± 0.10 Wm-2 online IRF."
But to be clear, these numbers are the changes in RF between 2015 and 2050, correct? Should these results be reported as "delta ERF", for example?
Would ERF for 2050 be estimated as 0.47 +/- 0.23 W m-2 (from IPCC AR6) plus 0.27±0.09 W m-2 (this study), to equal 0.74 +/- 0.31 W m-2 ?

Line 74
It would helpful to provide a little more background on the SSP3-7.0 scenario, which is generally described as being driven by "regional rivalry". While NOx and CH4 emissions go up in this scenario through 2050 (see figure 6.18 of IPCC AR6 WG-I), there are clear regional differences, especially for NOx (see figure 6.19 of IPCC AR6 WG-I). For example, NOx emissions decrease quite strongly by 2050 in North America, Europe, Russia, Central Asia, and Pacific OECD.

The information in Figure 2 is a very handy reference, but I think it would work better as a table so you can list important items for each estimate such as: TOA vs tropopause, ERF vs IRF, starting and ending year. There seems to be an error in the mid-point of ozone RF for IPCC AR 2, which is plotted as 0.5 W m-2. However, Page 20 of IPCC AR2 states:
"Changes in tropospheric ozone have potentially important consequences for radiative forcing. The calculated global average radiative forcing due to the increased concentration since pre-industrial times is 0.4 (±0.2) Wm-2."

In terms of comparing the model output to observed ozone changes (1995 through 2021) a new paper by the HEGIFTOM Working Group will soon be available for open review on EGUsphere:

> Van Malderen, R., Z. Zang, K.-L. Chang, R. Bjorklund, O. R. Cooper, J. Liu, C. Vigouroux, E. Maillard Barras, I. Petropavlovskikh, T. Leblanc, V. Thouret, P. Wolff, P. Effertz, A. Gaudel, H.G.J. Smit, A. M. Thompson, R. M. Stauffer, D. E. Kollonige, D. Tarasick (2024), Global Ground-based Tropospheric Ozone Measurements: Regional tropospheric ozone column trends from the HEGIFTOM homogenized ground-based profile ozone datasets, submitted to ACP (TOAR-II Community Special Issue)

The analysis provides long-term ozone trends for many regions around the world, based on merged datasets (ozonesondes, lidar, IAGOS, FTIR). In my opinion, this is the best observation-based summary of global tropospheric ozone trends. The results are similar to the findings of IPCC AR6 (Gulev et al., 2021), except there is now evidence for a decrease of tropospheric column ozone in the Arctic, and there is a clear drop in ozone in 2020 and 2021 that coincides with the COVID-19 economic downturn.

Line 333
Please provide more information on the time-slice definition. Is this a 10-year time-slice centered on 2015 (i.e. averaged over 2010-2019)?

**References**

Gulev, S.K., P.W. Thorne, J. Ahn, F.J. Dentener, C.M. Domingues, S. Gerland, D. Gong, D.S. Kaufman, H.C. Nnamchi, J. Quaas, J.A. Rivera, S. Sathyendranath, S.L. Smith, B. Trewin, K. von Schuckmann, and R.S. Vose, 2021: Changing State of the Climate System. In Climate Change 2021: The Physical Science Basis. Contribution of Working Group I to the Sixth Assessment Report of the Intergovernmental Panel on Climate Change [Masson-Delmotte, V., P. Zhai, A. Pirani, S.L. Connors, C. Péan, S. Berger, N. Caud, Y. Chen, L. Goldfarb, M.I. Gomis, M. Huang, K. Leitzell, E. Lonnoy, J.B.R. Matthews, T.K. Maycock, T. Waterfield, O. Yelekçi, R. Yu, and B. Zhou (eds.)]. Cambridge University Press, Cambridge, United Kingdom and New York, NY, USA, pp. 287–422, doi:10.1017/9781009157896.004

---

## Author Response (AR1)

We thank Owen Cooper, Chris Smith and an anonymous reviewer for their useful comments that have helped improve the manuscript. To address these comments we have made substantial additions to the text, particularly to add discussion, figures and tables covering the radiative impacts of the ODS changes. The effective radiative forcing from the stratospheric ozone recovery is significantly larger than expected from previous studies and from calculations based on the traditional stratospheric-temperature adjusted RF. To further understand the physical processes behind the meteorological adjustments, we have added further discussion, figures and tables covering changes in humidity and surface albedo. We hope that these additions to the paper will make the importance of the scientific outcomes clearer to readers.

Owen Cooper

General comments:

The paper begins with an excellent review of ozone radiative forcing, which will be a very helpful reference for the scientific community (especially Figure 1).

Thank you.

Abstract.

Here the main findings on ozone RF are reported as: "We find robust changes in ozone due to future changes in ozone precursors and ODSs. These lead to a positive radiative forcing of 0.27±0.09 Wm-2 ERF, 0.24 ± 0.021 W m-2 offline SARF, 0.29 ± 0.10 Wm-2 online IRF." But to be clear, these numbers are the changes in RF between 2015 and 2050, correct? Should these results be reported as "delta ERF", for example? Would ERF for 2050 be estimated as 0.47 +/- 0.23 W m-2 (from IPCC AR6) plus 0.27±0.09 W m-2 (this study), to equal 0.74 +/- 0.31 W m-2 ?

We have now clarified that the forcings are 2015 to 2050

In the shared socioeconomic pathway SSP-3-7.0We find robust increases in ozone due to future increases in ozone precursors and decreases in ODSs.  leading to a  radiative forcing increase from 2015 to 2050 of 0.268 ±0.089 Wm$^{-2}$ ERF, 0.24 ± 0.0521 W m$^{-2}$  SARF, and 0.288 ± 0.10 Wm$^{-2}$  IRF.

In this study we do not report these as changes since 1750, but use 2015 as the baseline. Care would be needed in combining our results with those from IPCC AR6, so we do not recommend this in this paper.

Line 74

It would helpful to provide a little more background on the SSP3-7.0 scenario, which is generally described as being driven by "regional rivalry". While NOx and CH4 emissions go up in this scenario through 2050 (see figure 6.18 of IPCC AR6 WG-I), there are clear regional differences, especially for NOx (see figure 6.19 of IPCC AR6 WG-I). For example, NOx emissions decrease quite strongly by 2050 in North America, Europe, Russia, Central Asia, and Pacific OECD.

We have added more detail here:

This scenario is chosen as it has the largest increase in tropospheric ozone (Keeble et al., 2021; Turnock et al., 2020) through increases in methane, $NO_x$ and other ozone precursors - although note $NO_x$ emissions decrease in OECD countries (Szopa et al., 2021).

The information in Figure 2 is a very handy reference, but I think it would work better as a table so you can list important items for each estimate such as: TOA vs tropopause, ERF vs IRF, starting and ending year. There seems to be an error in the mid-point of ozone RF for IPCC AR 2, which is plotted as 0.5 W m-2.

However, Page 20 of IPCC AR2 states: "Changes in tropospheric ozone have potentially important consequences for radiative forcing. The calculated global average radiative forcing due to the increased concentration since pre-industrial times is 0.4 (±0.2) Wm-2."

Thank you for your suggestion, we have added the information in figure 2 as a table in the supplement.

| Report | Nominal years | Tropospheric ozone forcing | References |
|---|---|---|---|
| SAR | 1850 to 1990 | $0.4 \pm 0.2$ Wm$^{-2}$ | Hauglustaine et al. (1994) |
| TAR | 1850 to 1990 | $0.35 \pm 0.15$ Wm$^{-2}$ | Berntsen et al. (1997, 2000; Brasseur et al., 1998; Hauglustaine et al., 1998; Haywood et al., 1998; Kiehl et al., 1999; Lelieveld & Dentener, 2000; Roelofs et al., 1997; Stevenson et al., 1998; Van Dorland et al., 1997) |
| AR4 | 1850 to 2000 | $0.35$ [0.25, 0.65] Wm$^{-2}$ | (Gauss et al., 2006; Hauglustaine & Brasseur, 2001; Liao & Seinfeld, 2005; Mickley et al., 2001, 2004; Shindell et al., 2003, 2005; Wong et al., 2004) |
| AR5 | 1750 to 2010 | $0.4 \pm 0.2$ Wm$^{-2}$ | (Skeie et al., 2011; Søvde et al., 2011; Stevenson et al., 2013) |
| AR6 | 1750 to 2019 | $0.45 \pm 0.225$ Wm$^{-2}$ | (Skeie et al., 2020) |

**Table S1: Tropospheric ozone radiative forcing (calculated as SARF) from the second (SAR) to sixth IPCC Assessment Reports.**

The value for SAR has been corrected – thank you for spotting this.

In terms of comparing the model output to observed ozone changes (1995 through 2021) a new paper by the HEGIFTOM Working Group will soon be available for open review on EGUsphere: Van Malderen, R., Z. Zang, K.-L. Chang, R. Bjorklund, O. R. Cooper, J. Liu, C. Vigouroux, E. Maillard Barras, I. Petropavlovskikh, T. Leblanc, V. Thouret, P. Wolff, P. Effertz, A. Gaudel, H.G.J. Smit, A. M. Thompson, R. M. Stauffer, D. E. Kollonige, D. Tarasick (2024), Global Ground-based Tropospheric Ozone Measurements: Regional

tropospheric ozone column trends from the HEGIFTOM homogenized ground-based profile ozone datasets, submitted to ACP (TOAR-II Community Special Issue) The analysis provides long-term ozone trends for many regions around the world, based on merged datasets (ozonesondes, lidar, IAGOS, FTIR). In my opinion, this is the best observation-based summary of global tropospheric ozone trends. The results are similar to the findings of IPCC AR6 (Gulev et al., 2021), except there is now evidence for a decrease of tropospheric column ozone in the Arctic, and there is a clear drop in ozone in 2020 and 2021 that coincides with the COVID-19 economic downturn.

Thank you for drawing our attention to this dataset which will be useful for the community. As we do not compare against observed changes in our study we will not cite this paper here.

Line 333 Please provide more information on the time-slice definition. Is this a 10-year time-slice centered on 2015 (i.e. averaged over 2010-2019)?

We have now clarified that this is a continuous year 2015

The control experiment (called *pdClim-control*) is a time-slice simulation for a continuousthe year 2015

Chris Smith

This paper provides results from a multi-model experiment of ozone radiative forcing (using three metrics: instantaneous, stratospheric adjusted and effective) for 2050 relative to 2015 under the SSP3-7.0 scenario. This addresses an important gap in the literature that was not available during the IPCC Sixth Assessment on the likely future evolution of ozone radiative forcing, and will be a valuable resource to the community that will be referenced for years to come.

Thank you for your comment.

The title and abstract don't make it clear that only a single scenario is considered: we trust that the future will follow SSP3-7.0. Either call out SSP3-7.0 specifically or frame in more general terms: "Climate forcing due to future ozone changes in a high emissions scenario…" for example. Why was this scenario selected – presumably because it had the biggest signal? In the abstract, please state the 2050 relative to 2015 timeframe.

Since the focus of the paper is the metrics and methods rather than the specifics of the scenario we prefer not to lengthen the title. The key point of the paper is not the final radiative forcing number, but a comparison of the different methods to calculate radiative forcing.

We now explain in section 1 why this scenario is chosen.

This scenario is chosen as it has the largest increase in tropospheric ozone (Keeble et al., 2021; Turnock et al., 2020) through increases in methane, $NO_x$ and other ozone precursors - although note $NO_x$ emissions decrease in OECD countries (Szopa et al., 2021).

We now clarify the timeframe:

., These leading to a positive radiative forcing increase from 2015 to 2050 of 0.2687 ±0.0849 Wm$^{2-2}$ ERF, 0.244 ± 0.05721 W m$^{-2}$ offline SARF, and 0.2889 ± 0.101 Wm$^{-2}$ online IRF

Line 56: I think that tropospheric ERF is AR6 for 1750-2019 was +0.41 W m-2 and not +0.45 W m-2. Not that we mentioned it in the AR6 text, but I did do the trop/strat split. Does the Skeie et al. number include climate effects on ozone forcing? More generally, I am supposing that the experimental design here, using 2015 SSTs, does not include the climate effects on ozone either. SSP3-7.0 is a degree or more warmer in 2050 than in 2015 in many models so the effects are probably not insignificant.

AR6: " The contributions to total SARF in CMIP6 (Skeie et al., 2020) are 0.39 ± 0.07 and 0.02 ± 0.07 W m–2 for troposphere and stratosphere respectively … The dataset is extended over the entire historical period following Skeie et al. (2020), with a SARF for 1750–1850 of 0.03 W m–2 and for 2010–2018 of 0.03 W m–2" – so this gives 0.45 W m–2 for the troposphere.

The Skeie et al. number does include climate effects. The experimental design here does not. This could be important for the comparison with CMIP6 ozone in section4.2. We have added the caveat:

although the CMIP6 ensemble includes changes in climate which are excluded in the TOAR-RF simulations.

The section starting 1.1 on radiative forcing should possibly be promoted to a level-1 section (section 2). Possibly also section 1.2, but could sit under the radiative forcing header. I also thought that section 1.1 was a bit textbook and may not be required for this paper, but the coordinator of TOAR-II likes it, so it's probably a matter of taste and my familiarity with the topic.

Thank you for the suggestion. We have renumbered as suggested.

Line 270: "present day" in GISS means 2015? In all cases where "present day" is mentioned, please be specific on the year(s) (e.g. line 337).

"present day" has been changed to "2015" throughout

Line 337: where do these climatologies come from? Are they model-specific or centrally provided?

DMS concentrations climatologies are model-specific.

Other boundary conditions, such as ocean concentrations of dimethyl sulphide (DMS), etc. are also prescribed by each model as climatologies appropriate for the present day

Lines 348-349: "the models' respective radiation and cloud schemes...": will changes in aerosols (that are not fixed) affect clouds, which will affect the ERF?

The aerosols only affect the chemistry schemes. Connections between aerosols and radiation and aerosols and cloud microphysics have been turned off. This has been clarified:

The models' respective radiation and cloud microphysics schemes continue to see year-2015 atmospheric concentrations of greenhouse gases, aerosols and cloud condensation nuclei, except for ozone.

Line 473: 298.3 ± 8.3 DU – any observations to compare this to? Figure 6 has data from NIWA, perhaps this could be compared?

Lines 545-548: I'd also say the historical trend of CMIP6 models compared to the obs is quite good, even if biased high.

This comparison has been added to section 4.2.

The historical trend in CMIP6 ozone agrees well with that observed, Hhowever, bothlike the CMIP6 ensemble mean and, the TOAR-RF ensemble mean areis systematically biased high relative to observations (283.5±1.1 DU) by approximately 10 DU.

Figure 8: the 150 ppb ozone tropopause forcings agree quite well between models. Is this expected and/or worth a comment?

A comment to this effect has been added:

There is close agreement between models in the tropospheric forcing, but more model variability when including stratospheric changes as indicated by the large standard deviation across models in the total ozone radiative forcing .

Very minor, editorial things

Line 51: reference after full stop

Fixed

Line 79: write out equation on a new line

Fixed

Line 116: comma after full stop

Fixed

Line 310: Walters citation as author (year).

Fixed

Line 545: Bodeker Scientific doesn't show up in the references.

Fixed

Line 576: superscript -2

Fixed

Anonymous Reviewer 2

General Comments

The manuscript compares across different metrics and methods to estimate ozone radiative forcing by first presenting a synthesis of prior work and then calculating the radiative forcing from 2015 to 2050 using the current generation of Earth system models. The manuscript documents inconsistencies in approaches in prior work as well as unique configurations in some models that complicate a straightforward inter-model comparison. This detailed documentation is invaluable to the modeling community and is a critical piece for interpreting some simulations and possibly the next round of multi-model studies. It does, however, lead to a lengthy manuscript in which major conclusions may be missed. At the same time, providing some additional context to the abstract and conclusions may help a reader understand the importance of the work. Two general suggestions:

1. Articulate more clearly the key messages in the abstract/conclusions with some short synthesis statements that provide slightly broader context. For example, the authors may wish to consider the following questions: How much confidence is there in estimates of ozone forcing and the stratosphere versus troposphere contributions as reported here and in prior work? Given the emphasis on this prior work in the introduction, I expected the authors to conclude by comparing their results to that earlier work. Do the conclusions drawn here have implications for the interpretation of historical radiative forcing estimates for ozone? How important is ozone relative to other greenhouse gases? How do the emission trends driving the ozone forcing in this scenario compare with other future scenarios?

We have revised the conclusions to include the context that this calculation is larger than that assumed in the IPCC AR6 and would make ozone the second most important contributor to radiative forcing over the 2015-2050 period in this scenario. We also draw attention to the calculations that the ERF metric indicates much larger climate effects than SARF for stratospheric ozone recovery.

The abstract has been completely rewritten

**Abstract.**

[revised manuscript text omitted]

> 2. Justify the use of SSP3-7.0. For example, is this the scenario that regional emissions have followed most closely in the last decade or simply the one that all the models ran? What are NOx, NMVOC, and CO emission trends as well as ODS and methane in this scenario? Consider showing a plot with zonal mean 2050-2015 emission changes (and ODS) to highlight regional differences.

We have now explained that the choice of SSP3-7.0 is a pragmatic choice designed to give the largest ozone signal. The key point of the paper is not the final radiative forcing number, but a comparison of the different methods and metrics to calculate radiative forcing. For this reason we do not discuss the trends in the different precursors or the regional differences.

This scenario is chosen as it has the largest increase in tropospheric ozone (Keeble et al., 2021; Turnock et al., 2020) through increases in methane, NO$_x$ and other ozone precursors - although note NO$_x$ emissions decrease in OECD countries (Szopa et al., 2021).

1. The abstract should include the time frame and scenario over which the forcings are calculated.

We have now clarified that the forcing is over the period 2015 to 2050 and for the SSP3-7.0 scenario.

In the shared socioeconomic pathway SSP3-7.0 we find robust increases in ozone due to future increases in ozone precursors and decreases in ODSs.  lead to a  radiative forcing increase from 2015 to 2050 of 0.26 ±0.08 Wm$^{-2}$ ERF, 0.24 ± 0.05 W m$^{-2}$  SARF, and 0.28 ± 0.10 Wm$^{-2}$  IRF.

2. Please clarify what appears to be conflicting findings: Checa-Garcia et al. (2018) report that tropospheric ozone cools the lower stratosphere (line 117) but Figure 9g shows warming in the lower stratosphere in response to increasing tropospheric ozone (fODS).

Thank you for pointing this out. We now discuss in section 4.7 that the modelled temperature changes can differ from the FDH calculations, and have added a figure S9 to show this.

Figure S9a shows more detail on the stratospheric temperature profile change for one model (EMAC). Here it can be seen that the increased temperature occurs in the lower and middle stratosphere (roughly from 200 to 20 hPa), and also around 2 hPa. Temperature changes at higher altitudes have less impact on the net TOA radiative forcing as the density of the atmosphere decreases. In the *fODS-perturbation*, however, SARF is enhanced by 0.06 Wm$^{-2}$ compared to IRF due to the decreased stratospheric temperatures as diagnosed by the FDH calculations (see Fig. S9). Note in Fig. 9h (and Fig S9a) there is warming in the modelled temperatures in the upper troposphere and lower stratosphere even in the *fODS-perturbation* that is not captured in the  FDH calculations.

3. Pages 2-3: Clarify focus here is not only on tropospheric ozone.

We have clarified that changes in tropospheric and stratospheric ozone are quantified.

This study assesses ozone radiative forcing using a combination of definitions and methodologies as part of the Tropospheric Ozone Assessment Report Phase II (TOAR-II). It quantifies forcing due to changes in both tropospheric and stratospheric ozone.

In section 2 a discussion of past studies of stratospheric ozone forcing is added:

Forcing from stratospheric ozone changes is sensitive to the altitude of the change, with decreases in lower stratospheric ozone contributing a negative forcing and decreases in upper stratospheric ozone contributing a positive forcing (Skeie et al., 2020). Estimates of historical stratospheric ozone forcing have therefore been uncertain even in the sign. IPCC AR6 (Forster et al., 2021) references a forcing from historical stratospheric ozone changes of $0.02 \pm 0.07$ W m$^{-2}$ based on offline kernel SARF calculations in Skeie et al. (2020).

The ozone radiative forcing from historical ODS increases is more robustly negative as it excludes contributions from increasing ozone precursors and includes the impact of ozone depletion on upper tropospheric concentrations. IPCC AR5 (Myhre et al., 2013) quantified an ozone SARF attributed to ODSs of $-0.15 \pm 0.15$ Wm$^{-2}$ (where the uncertainty is 5-95% confidence limit) and AR6 (Szopa et al., 2021) a value of -0.16 Wm$^{-2}$ (no confidence limit provided). An ozone ERF of -0.04 ± 0.03 Wm$^{-2}$ due to the historical ODS increase  has been calculated from one model (Michou et al., 2019), but it is not clear that this can be directly compared to the SARF calculations.

4. Line 42: Include example of a non-cloud adjustment the first time this term is mentioned.

We now give water vapour and albedo changes as examples.

Non-cloud adjustments  due to water vapour and albedo changes are positive .

5. Figure 1: Clarify if this is for tropospheric ozone only?

We have clarified that this is for ozone changes up to 0.1 hPa

Figure 1: Radiative efficiencies (SARF) for ozone changes up to 0.1 hPa  in mWm$^{-2}$ per DU based on calculations in Skeie et al. (2020)

6. Section 2.1: Different information is provided for different models (number of chemical species, reactions given for some but not others). Consider a summary Table with consistent information provided for all models. This table could be main text and detailed text describing unique model aspects could move to supplement to shorten main text.

Thank you for the suggestion. We have now included such a table and moved the more detailed model descriptions into the Supplementary Material.

| Model Name | Model Type | Horizontal Resolution | No of Vertical levels (Model lid) | Chemistry scheme | No. of species | No. of reactions Gas-phase | Aqueous -phase | Hetero geneou s | Photolysis | Radiative transfer scheme | Reference |
|---|---|---|---|---|---|---|---|---|---|---|---|
| CESM2 | Earth System Model | 0.9°x1.25° | 32 (2.26 hPa) | MOZART-TS1 | 221 | 405 | | 17 (strat.) | 123 | RRTMG | Danabaso glu et al. (2020), Emmons et al. (2020) |

etc

7. Lines 341-342: Was stabilization achieved in all the runs?

We clarify that the spin ups were long enough for the models to stabilise.

This was found to be sufficient for stabilisation.

8. Section 2.4: mention here that IRF is also calculated (line 797 points to IRF kernels but only SARF kernels are introduced)

We now mention that an IRF kernel is used too.

For comparison with the online IRF calculations the IRF kernel is the sum of SW and LW for either clear sky or all sky.

9. Figure 6: Is the tropopause used for the observations consistent with the definition in the models? Clarify that the TOAR value is from this study.

The comparison is for total column ozone.

We have added clarification that the TOAR value is from this study.

The TOAR-RF multi-model mean (black diamond), and the inter-model spread expressed as ±1 standard deviation from this study is based on the models: CESM2, EMAC, GEOS-Chem, GFDL-ESM4, GISS-E2.1_FR, NorESM2, and UKESM1-0-LL.

10. Sections 3.3/3.4/3.5 and 4.3/4.4: Add a final summary sentence or short paragraph to convey the key message(s) emerging from the findings reported in each section?

We have added summary sentences to these sections.

4.3

In summary, the net all-sky multi-model mean ERF is similar to the IRF within the uncertainties due to a reduction in the LW forcing, but an increase in the SW forcing. The ERF is higher than the SARF for the one model that calculated it, particularly in the clear-sky.

4.4

There is close agreement between models in the tropospheric forcing, but more model variability when including stratospheric changes as indicated by the large standard deviation across models in the total ozone radiative forcing .

4.5

Overall the models show a robust decrease in cloud cover in response to increased ozone. This causes a significant increase in SW forcing and a significant decrease in LW forcing, with a small decrease in net forcing that varies across models contributing to the differences in how rapid adjustments enhance or reduce the ERF compared to IRF.

5.3

**5.3.3 Summary of clear-sky comparison**

Both experiments show a decrease in surface albedo, with a consequently larger SW clear-sky ERF than SARF. This has a proportionally greater effect for the fODS perturbation. In the LW clear-sky the ERF is larger than the kernel SARF for the standard perturbation. For the effect of ODSs (standard minus fODS), the LW clear-sky ERF even has a different sign. This sign change between LW clear-sky ERF and SARF is also the case for the EMAC model which diagnoses both online. Since the kernel LW IRF was found to be 40% greater than the models (Section 5.2) the true increase of ERF compared to the SARF could be even larger.

5.4

There is a suggestion that the cloud adjustment is negative for the ozone precursor changes, and positive for the ODS changes, but the uncertainties are large<s>and no systematic difference between the standard and fODS perturbations</s>. Although uncertain<s>Nevertheless</s>, the cloud adjustment (-0.1 to +0.05 Wm$^{-2}$) is a significant fraction of the total ERF (0.27±0.<s>04 </s>08 Wm$^{-2}$).

11. Lines 589-591: Given the different cloud fields, is this finding meaningful?

This has been clarified to explain that the meaningful finding is that the different cloud fields mean that the fluxes are noisy

For all-sky ERF, the fluxes  show considerable noise  due to cloud changes between the simulations (see Sect. 4.5). This demonstrates that even with nudging there is large variability in cloudiness.

12. Line 653. GEOS-Chem reads in the cloud fields, so rather than a zero response it's the same clouds being used as input.

Agreed, this has been removed from the text and the cell in the table has been changed to "N/A".

13. Lines 720-722. So tropospheric ozone precursors and ODS (&N2O?) contribute equally to ERF in 2050 in this scenario? Figure 12 might better illustrate this by using hatching as in Figure 8. Consider adding this point to the abstract.

Thank you for your suggestion on the bar chart. We have revised this to better illustrate the comparison.

[Figure]

We agree on the importance of the ODS contribution, and have added this point to the abstract.

This increase makes ozone the second largest contributor to future warming by 2050 in this scenario, approximately half of which is due to stratospheric ozone recovery and half due to tropospheric ozone precursors.

An additional simulation varying N2O has been performed by one model (UKESM1). The text describing N2O in section 4.7 has been changed to include these calculation.

 $N_2O$  concentrations increase from a -2015 value of 328 to 362 nmol mol$^{-1}$ by 2050  in the *fODS-perturbation* simulation following SSP3-7.0. The effects on ozone of this $N_2O$ increase have been tested in one model (UKESM1.0-LL). in the SSP3-7.0 scenario on ozone. In the stratosphere, $NO_x$ is produced from $N_2O$ and hence $N_2O$ is foreseen to be an important factor with respect to ozone depletion in the 21$^{st}$ century . Prescribing increasing $N_2O$ surface mixing ratios leads to an ozone reduction in the middle and upper stratosphere, which would counteract the increase of stratospheric ozone related to the expected decrease~~

in ODSs until 2050 (; results based on a different emission pathway), despite ozone loss due to NO$_x$ being less efficient when CO$_2$ and CH$_4$ concentrations increase is found to have a negligible effect on the total ozone column (-0.2 DU) due mostly to depletion in the upper stratosphere (Ffig. S10?). It does cause a positive radiative forcing (0.034 $\pm$ 0.043 Wm$^{-2}$ ERF, 0.029 Wm$^{-2}$ SARF) since ozone in the upper stratosphere has a negative radiative efficiency (Fig.

Technical Corrections

1. On Figure 5, is the topography used to determine where to plot as a function of pressure the same in the different panels? Around 40S and 90N there are some differences.

Thank you for pointing this out. There was an error in how topography was used, which has now been corrected.

2. Bodeker Scientific, 2024 is not in the bibliography

Fixed

Other changes:

A table of ozone column differences has been added to section 4.1

[revised manuscript text omitted]

The albedo changes are discussed in 5.3.1

By excluding cloud effects, the clear sky ERFs should compare to the kernel SARFs if the tropospheric non-cloud adjustments are small. The modelled clear-sky SW ERF correlates well with the RF (equal to SARF in SW) kernel calculations (see Fig. 15). he ERF calculation is consistently higher than the IRF (see Table 7). This is also the case using the double-call clear-sky SW IRF for the models that diagnose it. This  is expected from the decreases in surface albedo diagnosed in Section 4.6. For the fODS experiments the albedo adjustment explains the difference between the ERF and IRF such that the residual defined as ERF-IRF-adjustment is zero, suggesting that albedo changes explain the difference. In the standard experiments when decreases in ODSs are included the residual is negative, suggesting that there may be further negative adjustments in the SW as a response to stratospheric ozone recovery. The ozone recovery leads to an increase in stratospheric water vapour (Fig. 9), but it is not obvious why this would lead to a negative SW forcing adjustment.

In this study we define the ERF using fixed SSTs. If land temperatures were also fixed it is likely that the albedo adjustment would be substantially reduced. If an ERF defined using fixed SSTs and land temperatures were required, as in IPCC AR6 (Forster et al., 2021), then it may be necessary to subtract the albedo adjustment from the diagnosed fSST ERFs. This would be analogous to subtracting the land surface temperature adjustment as in Tang et al. (2019).

| | Clear-sky SW Double-call IRF | Clear-sky SW kernel IRF | Clear-sky SW ERF | Albedo adjustment | Residual |
|---|---|---|---|---|---|
| Standard | | $0.115 \pm 0.032$ | $0.131 \pm 0.042$ | $0.040 \pm 0.017$ | $-0.024 \pm 0.012$ |
| Standard[*] | $0.125 \pm 0.25$ | | $0.152 \pm 0.042$ | $0.036 \pm 0.020$ | $-0.010 \pm 0.012$ |
| fODS | | $0.034 \pm 0.007$ | $0.050 \pm 0.012$ | $0.017 \pm 0.10$ | $0.000 \pm 0.004$ |
| fODS[*] | $0.031 \pm 0.009$ | | $0.050 \pm 0.014$ | $0.016 \pm 0.12$ | $0.003 \pm 0.007$ |

[*] Only models that included double-call diagnostic of IRF.

**Table 7: Comparison of the difference between ERF and IRF with the albedo adjustment in the clear-sky SW. Two definitions of IRF are used, kernel and double-call. Only a subset of models included double-call diagnostics. The residual is defined as ERF - IRF - albedo adjustment.**

---

## Author Response (AR2)

Author responses

We thank Chris Smith and an anonymous reviewer for their help in improving this work.

**Chris Smith**

Abstract:
- delete "For the first time" right at the start.
- Hence ERF "may be" a more appropriate metric; could be stronger here, use "is"?

Line 175: partial *radiative* perturbation, rather than partial radiation perturbation?
Line 785: Hence using ERF we find "well" nearly three times the sensitivity – I think "well" should be deleted

These changes have been made

**Anonymous reviewer**

The authors have thoroughly addressed all my concerns. The key messages are now much clearer. This paper is an important contribution and timely for CMIP7/AR7. A few additional small edits the authors may wish to consider:
Figure 14 right panel typo on label (a) instead of (b). Use the same range for x- and y-axes?
line 912 in conclusions, specify that these are tropospheric ozone precursors?

These changes have been made